# SMARTraj$^2$: A Stable Multi-City Adaptive Method for Multi-View Spatio-Temporal Trajectory Representation Learning

**Tangwen Qian**[1,3]**, Junhe Li**[1,3]**, Yile Chen**[2,*]**, Gao Cong**[2]**, Zezhi Shao**[1,3]**,**
**Jun Zhang**[1,3]**, Tao Sun**[1,3,*]**, Fei Wang**[1,3]**, Yongjun Xu**[1,3]

[1]State Key Laboratory of AI Safety, Institute of Computing Technology, Chinese Academy of Sciences
[2]College of Computing and Data Science, Nanyang Technological University
[3]University of Chinese Academy of Sciences
{qiantangwen,shaozezhi,suntao,wangfei,xyj}@ict.ac.cn, sljhhy@gmail.com,
yile001@e.ntu.edu.sg, gaocong@ntu.edu.sg, zhangjun254@mails.ucas.ac.cn

## Abstract

Spatio-temporal trajectory representation learning plays a crucial role in various urban applications such as transportation systems, urban planning, and environmental monitoring. Existing methods can be divided into single-view and multi-view approaches, with the latter offering richer representations by integrating multiple sources of spatio-temporal data. However, these methods often struggle to generalize across diverse urban scenes due to multi-city structural heterogeneity, which arises from the disparities in road networks, grid layouts, and traffic regulations across cities, and the amplified seesaw phenomenon, where optimizing for one city, view, or task can degrade performance in others. These challenges hinder the deployment of trajectory learning models across multiple cities, limiting their real-world applicability. In this work, we propose SMARTraj$^2$, a novel stable multi-city adaptive method for multi-view spatio-temporal trajectory representation learning. Specifically, we introduce a feature disentanglement module to separate domain-invariant and domain-specific features, and a personalized gating mechanism to dynamically stabilize the contributions of different views and tasks. Our approach achieves superior generalization across heterogeneous urban scenes while maintaining robust performance across multiple downstream tasks. Extensive experiments on benchmark datasets demonstrate the effectiveness of SMARTraj$^2$ in enhancing cross-city generalization and outperforming state-of-the-art methods. See our project website at `https://github.com/GestaltCogTeam/SMARTraj`.

## 1 Introduction

Spatio-temporal trajectory representation learning is fundamental to a variety of urban applications, including intelligent transportation systems [52, 28], urban planning [9, 46], and environmental monitoring [18, 44]. The goal is to encode spatio-temporal data (e.g., GPS coordinates, road networks, timestamps, and points of interest) into representations that capture the underlying patterns of urban mobility, facilitating diverse downstream tasks such as anomaly detection [42, 41], clustering [43, 29], and trajectory forecasting [48, 35].

Current methods can be divided into two subcategories: single-view and multi-view approaches. Single-view methods leverage one specific type of spatial data, such as GPS trajectories [23, 20],

---

*Corresponding Authors: Yile Chen, Tao Sun.

39th Conference on Neural Information Processing Systems (NeurIPS 2025).

road network routes [14, 13], or points of interest (POI) sequences [40, 30]. Although these methods effectively capture patterns within respective modalities, their reliance on a single view inherently limits ability to model the complex and multi-faceted nature of urban mobility. In contrast, multi-view approaches [24, 33] aim to enhance representation richness by integrating multiple types of spatio-temporal data, enabling a more comprehensive understanding of mobility behaviors. However, these methods are often constrained to datasets from a single city, significantly limiting their generalization capability to other urban scenes with distinct characteristics. Generalization across cities is crucial, as urban scenes exhibit considerable diversity in geography, infrastructure, and human mobility patterns. Methods lacking the ability to generalize across cities struggle to maintain robustness in real-world applications, where models need to adapt to diverse and unseen urban contexts. Thus, achieving generalization across cities is essential for stable and transferable spatio-temporal trajectory representation learning.

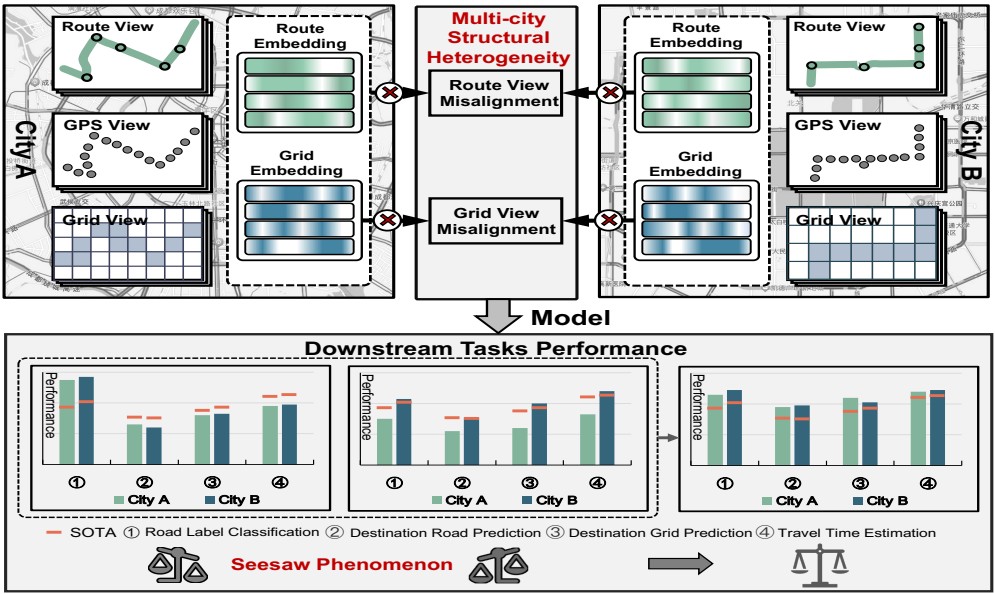

Figure 1: Existing multi-view spatio-temporal trajectory representation learning methods face critical challenges in generalizing across cities, which is crucial for real-world applications with diverse urban scenes: (1) multi-city structural heterogeneity, arising from disparities in urban layouts, and (2) the amplified seesaw phenomenon, where performance trade-offs between cities, views, and tasks are more pronounced.

To address this limitation, our goal is to enable spatio-temporal trajectory representation learning that generalizes across diverse urban scenes, which requires capturing universal spatio-temporal patterns shared across cities while also preserving city-specific characteristics to account for unique urban features. To achieve this, we propose a novel multi-city adaptive method that leverages multi-view spatio-temporal data to learn stable representations. However, realizing this goal introduces two critical challenges:

The first challenge is *multi-city structural heterogeneity*. Urban scenes exhibit significant disparities in structural layouts, such as road networks, grid partitions, and traffic regulations, which lead to distinct spatio-temporal patterns across cities. These differences make it difficult to develop a model that generalizes across diverse urban landscapes. Specifically, in multi-view settings, embedding spaces for corresponding views (e.g., route and grid views) across cities are inherently disjoint. For instance, road ID embeddings in one city operate in a different embedding space from those in another city, and grid ID embeddings often encode distinct spatial structures. This lack of alignment among embedding spaces prevents consistent representation learning, limiting the model's ability to generalize across cities.

The second challenge is the *amplified seesaw phenomenon*. In task-agnostic representation learning, balancing performance across multiple downstream tasks is critical. However, this balance is

inherently difficult in multi-view settings, where optimizing for one view often leads to performance degradation in others due to the heterogeneous nature of data across views. The challenge becomes even more complex in multi-city scenes. Specifically, multi-city, multi-view trajectory data introduces additional heterogeneity, as each city exhibits unique data distributions and structural characteristics. This amplifies the seesaw phenomenon, where improvements in performance for one city, view, or task may disproportionately degrade performance in others. Furthermore, achieving generalization while supporting multiple downstream tasks significantly complicates the learning process, making it challenging to maintain stable performance across cities, views, and tasks.

To tackle these challenges, we introduce SMARTraj[2], a **S**table **M**ulti-city **A**daptive method for **M**ulti-view spatio-temporal **Traj**ectory representation learning. To address the multi-city structural heterogeneity, we design a feature disentanglement module that separates domain-invariant and domain-specific features using orthogonality constraints, ensuring the model captures generalized spatio-temporal patterns while preserving city-specific characteristics. This disentanglement allows the model to adapt to new cities without losing critical information specific to the local urban structure. To mitigate the amplified seesaw phenomenon, we develop a personalized gating mechanism that dynamically adjusts the contributions of domain-invariant and domain-specific representations. The gating mechanism operates at both city-level and trajectory-level, adapting the contributions for different cities, views, and tasks. This ensures robust performance across cities while minimizing degradation in any specific view or task. By integrating them, SMARTraj[2] effectively stabilizes the trade-offs among cities, views, and tasks, enabling generalization across heterogeneous urban scenes.

The contributions of this work are summarized as follows:

- To the best of our knowledge, this is the first work to highlight the importance of multi-city, multi-view trajectory representation learning with a focus on generalization across diverse urban scenes. To improve this, we propose a novel method, SMARTraj[2], designed to learn stable representations from heterogeneous spatio-temporal data.

- We design a feature disentanglement module to separate domain-invariant and domain-specific representations, ensuring effective generalization while preserving city-specific characteristics. Additionally, a personalized gating mechanism is introduced to dynamically stabilize the contributions of different views and tasks, mitigating the amplified seesaw phenomenon in multi-city, multi-task settings.

- Extensive experiments demonstrate the superior performance of SMARTraj[2] compared to state-of-the-art methods, validating its stability in handling heterogeneous spatio-temporal data from various urban contexts.

## 2 Preliminaries

In this section, we introduce the fundamental concepts and formally define the problem addressed in this paper.

**Definition 1.** (**Trajectory**). A trajectory $T$ of length $|T|$ is a sequence of spatial and temporal data points, denoted as $T = \{(pos_i, t_i)\}_{i=1}^{|T|}$, where $pos_i$ represents the spatial location of the $i$-th sampled point (e.g., road segment ID, grid cell index, or exact latitude and longitude), and $t_i$ is the corresponding timestamp.

A trajectory can be represented in multiple ways, each capturing distinct spatial and contextual aspects of the underlying movement. Specifically, a multi-view trajectory representation integrates several spatial views, with each view offering unique insights into the underlying trajectory. These views include:

- GPS View $T^p$: A high-resolution view of the trajectory consisting of raw geographic coordinates, i.e., $pos_i = (lat_i, lon_i)$, where $lat_i$ and $lon_i$ are the exact latitude and longitude of the $i$-th point.

- Route View $T^r$: A structural view of the trajectory aligned with the road network, incorporating road segments and intersections, i.e., $pos_i = v_i$, where $v_i$ is the ID of the road segment associated with the $i$-th point.

- Grid View $T^g$: A macro-level view of the trajectory representing movements across a spatial grid, which may be augmented with semantic information, such as points of interest (POIs). Specifically, $pos_i = grid_i$, where $grid_i$ is the index of the grid cell containing the $i$-th point.

**Definition 2.** (**Multi-View Spatio-Temporal Trajectory Representation Learning**). Given a multi-view trajectory dataset $\mathcal{T} = \{(T_i^p, T_i^r, T_i^g)\}_{i=1}^{|\mathcal{T}|}$, the objective of multi-view spatio-temporal trajectory representation learning is to learn robust, task-agnostic representations for different views. These representations should generalize across various downstream tasks, such as road label classification, travel time estimation, and destination grid prediction.

Building on the previous definitions, we formally define the problem addressed in this paper.

**Problem Statement**. Let $D = \{D_1, D_2, \cdots, D_{|D|}\}$ be a dataset consisting of multi-view trajectories collected from multiple cities. For each city $k$, the dataset $D_k$ is composed of multi-view trajectory dataset $D_k = \{\mathcal{T}_i\}_{i=1}^{|D_k|}$, where each trajectory $\mathcal{T}_i = (T_i^p, T_i^r, T_i^g)$ is a tuple containing the three views, GPS view $T_i^p$, route view $T_i^r$, and grid view $T_i^g$. The goal is to learn transferable trajectory representations that integrate spatial information from these different views, while adapting to the unique characteristics of each city. The learned representations should enable stable performance across various downstream tasks by generalizing effectively across diverse urban scenes.

## 3 Method

In this section, we introduce the architecture of SMARTraj$^2$, detailing its core components: the feature disentanglement module and the personalized gating mechanism, designed to address the challenges posed by multi-city structural heterogeneity and mitigate the amplified seesaw phenomenon often observed in urban trajectory data.

### 3.1 Overview

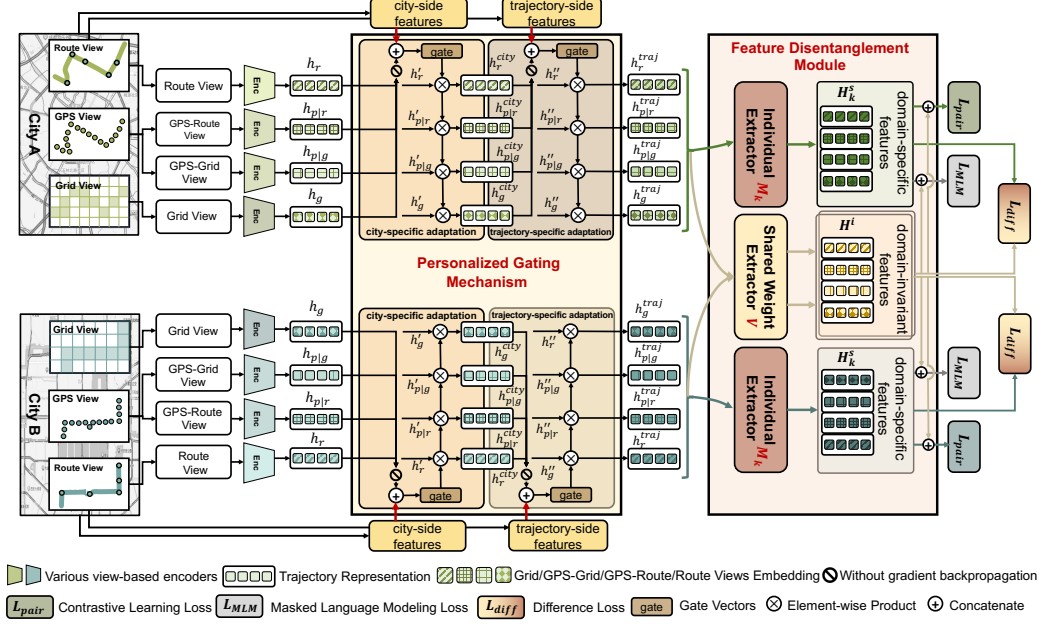

Figure 2: An overview of SMARTraj$^2$, consisting of two main components: the feature disentanglement module and the personalized gating mechanism.

As illustrated in Fig. 2, SMARTraj$^2$ comprises two key modules: (1) the feature disentanglement module separates domain-invariant and domain-specific features, utilizing orthogonality constraints to ensure the model captures generalized spatio-temporal patterns while maintaining city-specific characteristics. This disentanglement allows for flexible adaptation to new cities without losing

essential local urban information. (2) the personalized gating mechanism dynamically adjusts the contributions of domain-invariant and domain-specific features at both the city-level and trajectory-level. This ensures robust performance across cities while minimizing performance degradation in any specific view or task. This mechanism stabilizes the trade-offs between cities, views, and tasks, thereby enabling generalization across heterogeneous urban scenes.

### 3.2  Feature Disentanglement Module

The feature disentanglement module focuses on capturing spatial and temporal dependencies across multiple trajectory views, leveraging specialized encoders for each view. This module is pivotal for disentangling domain-invariant and domain-specific features, thus addressing the multi-city structural heterogeneity inherent in urban trajectory data.

For the GPS view, trajectories are first hierarchically segmented into sub-trajectories corresponding to road segments or grid cells. The GPS encoder processes these segments as follows:

$$
\begin{aligned}
h_{p|r} &= GPSEncoder(\mathcal{T}^{p|r}, B^{p|r}) \\
h_{p|g} &= GPSEncoder(\mathcal{T}^{p|g}, B^{p|g})
\end{aligned}
\tag{1}
$$

where $h_{p|r}$ and $h_{p|g}$ represent the encoded GPS trajectories aligned with road segments and grid cells, respectively. The hierarchical structure involves encoding individual GPS points using a bidirectional GRU, followed by encoding the resulting sub-trajectories, creating a two-level architecture. The binary assignment matrices $B^{p|r}$ and $B^{p|g}$ indicate associations between GPS points and their corresponding road segments or grid cells.

The route view incorporates spatial and temporal characteristics constrained by the road network and traffic dynamics. A graph attention network updates road segment spatial embeddings $z_v$ based on observed trajectories. Temporal features for each road segment $t_v$ combining discrete (e.g., day of the week) and continuous (e.g., travel time) variables, are added to form the final segment representation $r_v = z_v + t_v$. A transformer-based architecture is then used to encode the dependencies between road segments.

$$
h_r = RouteEncoder(\mathcal{T}^r, r_v)
\tag{2}
$$

where $h_r$ denotes the encoded route representation.

The grid view captures spatial relationships between grid cells, integrating semantic information from Points of Interest (POIs) to reflect the functional attributes of different areas. A transformer-based encoder models these dependencies:

$$
h_g = GridEncoder(\mathcal{T}^g, s(\mathcal{T}^g))
\tag{3}
$$

where $h_g$ is the grid trajectory representation, and $s(\mathcal{T}^g)$ represents the semantic embedding computed as a weighted sum of POI category embeddings within the grid cells.

Following the principles of transfer learning [16, 12, 5], we extract domain-invariant features $H^i$ using a shared-weight extractor $\mathcal{V}$:

$$
H^i = \mathcal{V}(h_p, h_r, h_g)
\tag{4}
$$

This module captures features common across cities, fostering robustness in the model's generalization capabilities.

Apart from city invariant features, in the context of multi-city, the incorporation of city-specific features significantly contributes to enhancing the performance and adaptability of models across different cities [32, 54, 37]. While domain-invariant features capture the shared underlying patterns and knowledge among various domains, domain-specific features account for the unique characteristics specific to individual domains. Consequently, the extraction of specific features from trajectories in different cities becomes of paramount importance. To complement this, domain-specific features $H_k^s$ are captured for each city using individual extractors $\mathcal{M}_k$, which are trained separately for each city:

$$
H_k^s = \mathcal{M}_k(h_p, h_r, h_g), 1 \leq k \leq |D|
\tag{5}
$$

There exists an orthogonality constraint between domain-specific and domain-invariant features [55, 45, 3]. Building upon this constraint, our method integrates the difference loss $\mathcal{L}_{diff}$ via a soft subspace orthogonality constraint between domain-specific $H_k^s$ and domain-invariant $H^i$

representations of each city to encourage a clear separation between the features related to specific domains and the features shared across domains.

$$\mathcal{L}_{diff} = \sum_{k=1}^{|D|} \|H^{i^\top} H_k^s\|_F^2 \tag{6}$$

where $\| \cdot \|_F^2$ denotes the squared Frobenius norm.

### 3.3 Personalized Gating Mechanism

The personalized gating mechanism allows the model to inject city-level and trajectory-level specific information into the embedding process, enabling dynamic adaptation.

For city-level adaptation, we generate city-specific gate scores based on city-side features, such as average trajectory length and speed, using a two-layer feedforward network. Specifically, we inject city-level specific personalized prior information into the embedding by using city-side features $E(\boldsymbol{F}_{city})$ (e.g., trajectory speed and statistics of trajectory length) as the input. We concatenate $h_r$ with the input $E(\boldsymbol{F}_d)$, but without using gradient backpropagation, denoted as $\overline{\nabla}(\cdot)$:

$$h_r' = \max(0, \boldsymbol{W}_{city}(\overline{\nabla}(h_r)\|E(\boldsymbol{F}_{city})) + \boldsymbol{b}_{city}) \tag{7}$$

here, $h_r'$ represents the intermediate feature vector. $(\cdot\|\cdot)$ denotes the concatenate operation. After crossing features with various prior information, we customize the generation of gate scores through a sigmoid function and modulate the embeddings:

$$\boldsymbol{\delta}_r^{city} = \gamma \cdot \sigma(\boldsymbol{W}_{city}' h_r' + \boldsymbol{b}_{city}') \tag{8}$$

$\sigma(\cdot)$ denotes the sigmoid function, which is used to generate gate vectors $\boldsymbol{\delta}_r$ and limits the output to $[0, \gamma]$. $\gamma$ is the scaling factor that is set as 2. We perform the personalized transformation on embedding $h_r$ without changing the original embedding layer, aligning features with different importance for different cities. The gate scores modulate the embeddings:

$$h_r^{city} = \boldsymbol{\delta}_r^{city} \odot h_r \tag{9}$$

where $\odot$ denotes the element-wise product.

For trajectory-level adaptation, we utilize features such as POI semantics at the start and end points to further personalize the network layers. Trajectory-specific gate scores are computed by concatenating the city-level gate scores with trajectory features:

$$h_r'' = \max(0, \boldsymbol{W}_{traj}(\overline{\nabla}(\boldsymbol{h}_r^{city})\|E(\boldsymbol{F}_{traj})) + \boldsymbol{b}_{traj}) \tag{10}$$

we modify all DNN layer parameters by using trajectory-side features $E(\boldsymbol{F}_{traj})$ (e.g., the starting and ending points' POI semantic feature). We concat the $\boldsymbol{h}_r^{city}$ with the trajectory-side features $E(\boldsymbol{F}_{traj})$ as the input. To avoid affecting the embedding updated in $\boldsymbol{h}_r^{city}$, we perform the operation of stop gradient $\overline{\nabla}(\cdot)$ on $\boldsymbol{h}_r^{city}$.

$$\boldsymbol{\delta}_r^{traj} = \gamma \cdot \sigma(\boldsymbol{W}_{traj}' h_r'' + \boldsymbol{b}_{traj}') \tag{11}$$

We use the element-wise product to double and squash the hidden contributions in layer of the DNN, fully personalize DNN parameters, balancing with different sparsity for different trajectories, formulated as follows. This gate are applied to adjust DNN parameters, ensuring tailored transformations:

$$h_r^{traj} = \boldsymbol{\delta}_r^{traj} \odot \boldsymbol{h}_r^{city} \tag{12}$$

To further enhance representation learning, we employ two additional loss functions: the masked language modeling loss $\mathcal{L}_{MLM}$, and the contrastive learning loss $\mathcal{L}_{pair}$.

The masked language modeling loss randomly masks portions of the trajectory data, forcing the model to predict masked elements and thereby learn generalized representations of the trajectory views. Formally, this loss $\mathcal{L}_{MLM}$ is defined as the negative log-likelihood of correctly predicting the masked tokens:

$$\mathcal{L}_{MLM} = \mathbb{E}(\mathcal{T}_m)[-\log P(\mathcal{T}_m \mid \mathcal{T}_{\backslash m})] \tag{13}$$

The contrastive learning loss distinguishes positive trajectory pairs that represent the same underlying trajectories across different views, from negative pairs that are randomly sampled.

$$\mathcal{L}_{pair} = \sum_{(i,j)\in P} \log \frac{\exp(sim(H^i, H^j))}{\sum_{(i,k)\in N} \exp(sim(H^i, H^k))} \qquad (14)$$

The overall training objective integrates multiple loss functions to stabilize different components:

$$\mathcal{L}_{total} = w_1 \mathcal{L}_{diff} + w_2 \mathcal{L}_{MLM} + w_3 \mathcal{L}_{pair} \qquad (15)$$

where $\mathcal{L}_{diff}$, $\mathcal{L}_{MLM}$, and $\mathcal{L}_{pair}$ represent the difference loss (Eq.6), the masked language modeling loss (Eq.13), and the contrastive learning loss (Eq.14). $w_1$, $w_2$, and $w_3$ are hyperparameters introduced to adjust relative weights between them. This loss function ensures that the model learns effective representations, promoting both domain-invariant and domain-specific adaptability.

# 4 Experiments

To evaluate the performance of SMARTraj[2], we conduct extensive experiments to answer the following research questions:

- **RQ1**: How does SMARTraj[2] compare to state-of-the-art trajectory representation learning models? (Sec. 4.2)

- **RQ2**: What is the impact of pre-training on the effectiveness of SMARTraj[2]? (Sec. 4.3)

- **RQ3**: How does each component of SMARTraj[2] contribute to its overall performance? (Sec. 4.4)

- **RQ4**: How do hyperparameters influence the performance of SMARTraj[2]? (Sec. 4.5)

## 4.1 Experimental Setup

### 4.1.1 Datasets

We conduct experiments on two real-world trajectory datasets from Chengdu and Xi'an, provided by DiDi Chuxing[1], along with road network data from OpenStreetMap[2]. These datasets contain GPS trajectories collected over 15 consecutive days in the central urban areas of both cities. The first 13 days are used for training, the 14th for validation, and the 15th for testing.

### 4.1.2 Downstream Tasks and Evaluation Metrics

To assess the generalization and effectiveness of the trajectory embeddings, we evaluate performance across four distinct downstream tasks, consistent with prior studies [24, 51, 22, 25]. These tasks encompass both fine-grained (e.g., destination grid prediction) and coarse-grained (e.g., road label classification) aspects of trajectory modeling.

- Road Label Classification: classifies road segments into four categories: Primary, Secondary, Tertiary, and Residential. Performance is measured using Micro-F1 and Macro-F1 scores.

- Travel Time Estimation: predicts the travel time of trajectories across all views. Performance is evaluated using Mean Absolute Error (MAE) and Root Mean Square Error (RMSE) .

- Destination Road Prediction: predicts the destination road segment of a trajectory based on its embedding derived from the route view. Performance is evaluated using top-k accuracy metrics ($Acc@k$), which measure the proportion of times the correct destination road appears within the top-k predictions.

- Destination Grid Prediction: predicts the destination grid cell of a trajectory, using its embedding derived from the grid view. Performance is evaluated using $Acc@k$.

Due to space constraints, further experimental setup details are provided in Appendix B.1.

Table 1: Overall Performance in Xi'an.

| Method | Road Label Micro-F1 / Macro-F1 | Travel Time MAE / RMSE | Destination Road Acc@1 / Acc@5 | Destination Grid Acc@1 / Acc@5 |
|---|---|---|---|---|
| Random | 0.4680 / 0.3087 | 120.9861 / 153.4056 | 0.6502 / 0.8035 | ✗ / ✗ |
| Word2vec | 0.6525 / 0.6267 | 89.5472‡ / 122.3465 | 0.6415 / 0.8139 | ✗ / ✗ |
| Node2vec | 0.4387 / 0.2938 | 91.5226 / 124.4122 | 0.6809 / 0.8116 | ✗ / ✗ |
| Transformer | 0.4305 / 0.3645 | 91.3093 / 124.1358 | 0.6662 / 0.8426 | ✗ / ✗ |
| BERT | 0.6780 / 0.6251 | 90.2442 / 123.2867 | 0.6898 / 0.851 | ✗ / ✗ |
| Toast | 0.6251 / 0.6182 | 88.0744 / 116.7965 | 0.6743 / 0.7362 | ✗ / ✗ |
| JCLRNT | 0.7445 / 0.7199 | 92.3900 / 125.5088 | 0.5504 / 0.7442 | ✗ / ✗ |
| START | 0.4413 / 0.3575 | 118.0605 / 162.0801 | 0.6778 / 0.8072 | ✗ / ✗ |
| JGRM | 0.7745‡ / 0.7622‡ | 87.8708‡ / 119.9921‡ | **0.7742** / 0.9063† | ✗ / ✗ |
| MVTraj | 0.8290† / 0.8159† | 54.9044† / 85.3847† | 0.6904‡ / 0.855‡ | 0.6630† / 0.8154† |
| SMARTraj[2] | **0.8407** / **0.8298** | **35.0689** / **60.9156** | 0.7409† / **0.9069** | **0.6675** / **0.8392** |

\* **Bold** denotes the best result, † and ‡ denotes the second and third best result.

## 4.2 Performance Comparison

Table 1 and Table 4 compare the performance of SMARTraj[2] against various baseline methods on the Chengdu and Xi'an datasets across multiple trajectory representation tasks.

SMARTraj[2] consistently outperforms state-of-the-art baselines across evaluated tasks. Specifically, in Chengdu, SMARTraj[2] reduces MAE by 29.30% and RMSE by 23.75% in travel time estimation, compared to MVTraj. A similar improvement is observed in Xi'an, highlighting the model's ability to generalize across cities with distinct mobility patterns.

Furthermore, unlike baselines that train independently on data from a single city per experiment, SMARTraj[2] is trained across multiple cities simultaneously, overcoming the structural heterogeneity that limits baseline methods. This enables SMARTraj[2] to remain stable in new urban environments without requiring retraining from scratch. Additionally, the personalized gating mechanism dynamically adjusts feature contributions across cities, alleviating the seesaw phenomenon, and ensuring consistent and stable performance across diverse urban settings.

## 4.3 Model Analysis

We assess the impact of pre-training on model performance across two datasets: Chengdu (Fig. 3) and Xi'an (Fig. 5). Specifically, we evaluate two distinct training paradigms:

- Pre-train: This corresponds to the original SMARTraj[2], where self-supervised objectives are first used to pre-train the trajectory encoder. The model is then fine-tuned on either the travel time estimation task or the destination road prediction task .

- No Pre-train: This variant is trained in an end-to-end manner, where both the trajectory encoder and the prediction head are randomly initialized and jointly optimized from scratch using supervised task-specific labels.

We observe that our results consistently demonstrate that pre-training significantly improves model effectiveness compared to training from scratch. We observe that pre-training not only accelerates convergence but also reduces dependency on labeled data during fine-tuning, making the model more robust to data scarcity.

## 4.4 Ablation Study

We conduct a comprehensive ablation study to evaluate the contribution of key components in our method. Specifically, we examine the following model variants:

- w/o diff loss: Removes the difference loss $\mathcal{L}_{diff}$, which applies soft orthogonality constraints to disentangle domain-invariant and domain-specific features.

---

[1] https://outreach.didichuxing.com/

[2] https://www.openstreetmap.org/

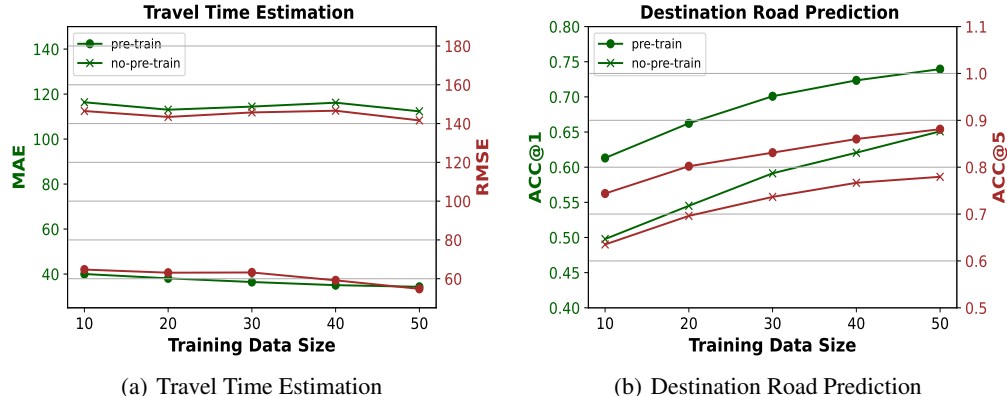

(a) Travel Time Estimation  (b) Destination Road Prediction

Figure 3: Effect of Pre-training in Chengdu.

Table 2: Ablation Study in Xi'an.

| Method | Road Label Micro-F1 / Macro-F1 | Travel Time MAE / RMSE | Destination Road Acc@1 / Acc@5 | Destination Grid Acc@1 / Acc@5 |
|---|---|---|---|---|
| SMARTraj[2] | 0.8407 / 0.8298 | 35.0689 / 60.9156 | 0.7409 / 0.9069 | 0.6675 / 0.8392 |
| w/o diff loss | 0.8500 / 0.8366 | 44.8105 / 73.8862 | 0.6025 / 0.8125 | 0.4866 / 0.7119 |
| w/o gating | 0.8387 / 0.8279 | 40.3469 / 68.6216 | 0.6787 / 0.8478 | 0.5244 / 0.7200 |
| w/o grid | 0.8233 / 0.8186 | 72.6226 / 105.6123 | 0.6604 / 0.8402 | ✕ / ✕ |
| w/o GPS | 0.7987 / 0.7832 | 73.2965 / 106.1142 | 0.5446 / 0.7667 | 0.4110 / 0.6351 |
| w/o route | ✕ / ✕ | 74.5902 / 106.7897 | 0.5924 / 0.8049 | 0.4311 / 0.6636 |
| w/o invariant+specific | 0.7415 / 0.7268 | 56.5380 / 71.5127 | 0.4770 / 0.7001 | 0.2556 / 0.5315 |
| w/o gating+specific | 0.7637 / 0.7574 | 54.8681 / 84.8157 | 0.6300 / 0.8294 | 0.5466 / 0.7636 |

- w/o gating: Excludes the personalized gating mechanism, which injects city-level and trajectory-level specific information, enabling adaptive feature modulation.

As shown in Table 2 and Table 5, both components contribute significantly to model performance. Removing the difference loss (w/o diff loss) results in severe performance degradation, demonstrating that $\mathcal{L}_{diff}$ effectively separates domain-invariant and domain-specific information, enabling stable spatio-temporal modeling across cities. Excluding the personalized gating mechanism (w/o gating) also leads to notable performance drops, suggesting that gating plays a key role in dynamically balancing the contributions of city-specific and global features.

Moreover, we conduct an ablation study to evaluate the model under limited-view settings. The results below demonstrate that our method maintains reasonable performance even when only GPS data is available.

### 4.5 Parameter Sensitivity

We perform a sensitivity analysis on key hyperparameters: the scaling factor $\gamma$ (defined in Eq(8)) and the weight ratio between $w_1$ and $w_2$ (defined in Eq(15)), where $w_1$ and $w_2$ are the weights for the difference loss and masked language modeling loss, respectively. Results for travel time estimation in Xi'an are presented in Fig. 4. Due to space limitations, additional results for destination road prediction are provided in Appendix B.2.4, with consistent trends.

Fig. 4(a) illustrates that $\gamma$ achieves optimal performance when set to 2. This factor controls the output range of gate scores, which modulate the embeddings for different cities and trajectories. Specifically, $\gamma = 2$ effectively balances the modulation by restricting the gate values to the range $[0, 2]$ and centering them around 1. Fig. 4(b) demonstrates that the best performance is attained when the weight ratio $w_1 : w_2 = 1$. Ratios $w_1 : w_2 < 1$ weaken the orthogonality constraint, while ratios $w_1 : w_2 > 1$ disrupt the balance between loss components, both leading to performance degradation.

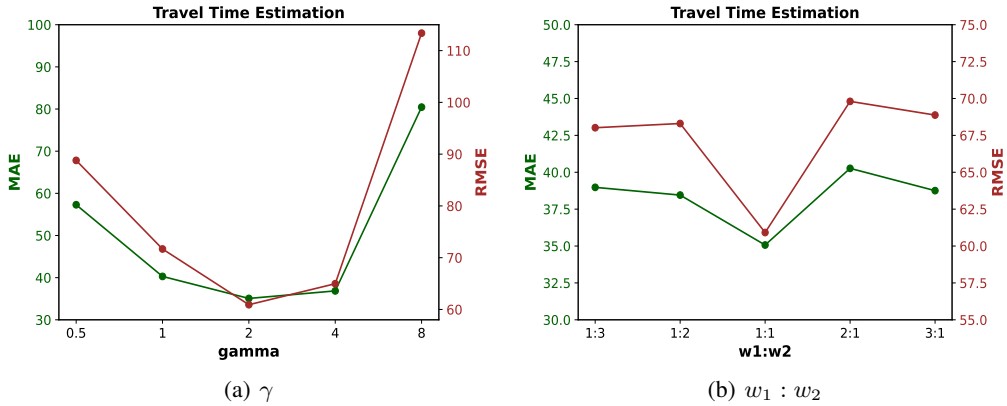

(a) $\gamma$         (b) $w_1 : w_2$

Figure 4: Parameter Sensitivity Analysis on Travel Time Estimation in Xi'an.

# 5 Conclusion

In this paper, we propose SMARTraj[2], a novel method for multi-view spatio-temporal trajectory representation learning that addresses the critical limitation of generalization across diverse urban scenes. We identified two key challenges that hinder the performance of existing approaches: multi-city structural heterogeneity, where cities exhibit significant differences in spatio-temporal patterns, and the amplified seesaw phenomenon, which arises when balancing performance across multiple cities, views, and tasks. To overcome these challenges, SMARTraj[2] leverages a feature disentanglement module to separate domain-invariant and domain-specific features, enabling the model to capture generalized spatio-temporal patterns while preserving city-specific characteristics. Additionally, a personalized gating mechanism dynamically adjusts the contributions of these features, mitigating the seesaw effect and ensuring stable performance across diverse urban scenes. Extensive experiments on real-world datasets show that SMARTraj[2] consistently outperforms state-of-the-art methods, demonstrating its ability to generalize effectively across cities with distinct mobility patterns, and proving its robustness in real-world applications.

**Limitation and Future Work.** Although the proposed framework demonstrates strong adaptability and performance across two representative cities (Chengdu and Xi'an), several limitations remain. The model depends on multi-view urban data, and its performance may be influenced by missing modalities or inconsistent data quality across cities. Future work will focus on developing robust data fusion and modality-adaptive mechanisms, extending experiments to larger city networks (e.g., 10+ cities) to validate scalability, and exploring efficient training and inference strategies such as parameter sharing, model compression, and distributed learning.

**Social Impact.** The proposed framework has potential societal benefits in improving urban mobility management, traffic forecasting, and resource allocation. However, it also raises important ethical and privacy concerns, particularly when dealing with trajectory or location-based data. Individual mobility traces may reveal sensitive information about users' habits or identities, posing privacy risks if mishandled. To mitigate these risks, strict data anonymization, aggregation, and de-identification procedures should be enforced before model training. Additionally, data access should comply with local data protection regulations and institutional review protocols.

## Acknowledgments and Disclosure of Funding

This work is supported by NSFC No. 62502499, NSFC No. 62372430, NSFC No. 62502505, the Youth Innovation Promotion Association CAS No.2023112, the Postdoctoral Fellowship Program of CPSF under Grant Number GZC20241758, the Postdoctoral Fellowship Program of CPSF under Grant Number GZC20251078, and the China Postdoctoral Science Foundation No.2025M771542.

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

# A   Related Work

In this section, we first review existing research on trajectory representation learning, followed by an exploration of transfer learning approaches in spatio-temporal domains.

## A.1   Spatio-Temporal Trajectory Representation Learning

Spatio-temporal trajectory representation learning has garnered substantial attention due to its relevance in various applications, including intelligent transportation systems [31, 28], urban planning [47, 46], and environmental monitoring [18, 44]. Existing approaches can be categorized into task-specific and task-agnostic methods.

Task-specific methods are designed to optimize performance on a particular downstream task, such as anomaly detection [42, 41], clustering [43, 29], and trajectory forecasting [34, 35]. These methods typically optimize trajectory encoders with task-specific objectives, resulting in high performance for the targeted application. However, their narrow focus limits their ability to generalize across diverse tasks and makes them less efficient in real-world multi-task scenarios [1, 17].

Task-agnostic methods, on the other hand, aim to learn generalized representations that can be applied across various downstream tasks. Many of these methods employ self-supervised learning techniques [22, 8] to enable flexible generalization. These methods are further divided into single-view and multi-view approaches. Single-view methods rely on a single spatial aspect of trajectory data, such as raw GPS coordinates [21, 56], road network routes [50, 51], or POI sequences [40, 6]. While these methods effectively capture patterns within the chosen view, they often fail to model the full complexity of spatio-temporal data, particularly in diverse urban scenes. Multi-view methods [33, 24] attempt to address this limitation by integrating multiple data sources, offering a richer and more comprehensive understanding of movement patterns.

Despite these advancements, existing multi-view approaches are constrained to datasets from a single city, significantly limiting their generalization capability to other urban scenes with distinct characteristics. Our proposed SMARTraj[2] method addresses this critical gap by focusing on generalization across cities. We disentangle domain-invariant and domain-specific representations, ensuring effective generalization while preserving city-specific characteristics. Additionally, a personalized gating mechanism is operated at both city-level and trajectory-level to dynamically stabilize the contributions of different views and tasks, mitigating the amplified seesaw phenomenon.

## A.2   Transfer Learning for Spatio-Temporal Trajectories

The concept of transfer learning has seen tremendous success in fields such as natural language processing [4, 10, 38] and computer vision [11, 36, 2]. In the context of spatio-temporal trajectory analysis, transfer learning has recently emerged as a promising direction to handle the complexities inherent in this domain. These challenges include irregular sampling intervals, spatial heterogeneity, and intricate temporal dependencies that require sophisticated models to generalize effectively across diverse datasets.

Several studies have explored the application of transfer learning to spatio-temporal trajectory data. For example, [56] maintains robust representation capabilities for GPS data with varying qualities, effectively handling issues like noise, missing values, and inconsistent sampling rates. [53] partitions cities into non-overlapping areas and trains across multiple cities to achieve universal spatio-temporal prediction, excelling in few-shot and zero-shot tasks. [27] presents a generalizable deep learning model for weather and climate science that can handle heterogeneous datasets across different spatio-temporal dimensions.

However, existing transfer learning methods predominantly focus on single-view trajectory data (e.g., GPS or grid data), making them less effective for more complex scenarios involving multi-view data. Our work addresses this gap by introducing a solution that operates in multi-city, multi-view, and multi-task settings. The incorporation of multiple views exacerbates the challenges of transfer learning, particularly with respect to the seesaw phenomenon, where performance trade-offs between different views and tasks become more pronounced. Our method mitigates this issue by dynamically adjusting the contribution of shared and domain-specific features using a personalized gating mechanism, thereby stabilizing performance across cities, views, and tasks.

# B Technical Appendices and Supplementary Material

## B.1 Experimental Setting

### B.1.1 Details of Datasets

To facilitate a clearer comparison between the two datasets, we present key statistics in Table 3. These statistics help highlight the differences in composition between the Chengdu and Xi'an datasets, ensuring a comprehensive understanding of their characteristics.

For comparative purposes, our preprocessing steps are aligned with those used in prior studies [24, 33, 7, 25]. Specifically, to obtain route view trajectories $T^r$, we apply a map-matching algorithm [49] to convert raw GPS data into sequences of road segments, thereby producing road-network-constrained trajectories in the route view. Additionally, to ensure the relevance of the data, we preprocess the road network by filtering out segments that are not traversed by any trajectory. To obtain grid view trajectories $T^g$, we utilize Points of Interest (POI) data, collected from an external source[3], to enhance the semantic information of grid cells. Each grid cell's semantic representation is normalized based on the POIs it contains.

We further filter the trajectories to ensure data quality and consistency across experiments. Trajectories must contain between 10 and 100 road segments, 10 and 100 grid cells, or 10 and 256 GPS points. Any trajectories that do not meet these criteria are excluded from the dataset.

Each dataset consists of 13 distinct categories of points-of-interest (POI), representing a diverse range of urban functions. These categories include Dining, Scenery, Public Facilities, Shopping, Transportation, Education, Finance, Residential, Life Services, Sports, Healthcare, Government Offices, and Accommodation Services. These categories are crucial for enriching the semantic features of grid cells and offer a comprehensive representation of the urban scene.

Table 3: Details of the Datasets

| Datasets | Chengdu | Xi'an |
|---|---|---|
| Time Interval | 3.07 | 3.10 |
| Number of Nodes | 6450 | 4996 |
| Avg. Node Degree | 5.08 | 4.75 |
| Number of Edges | 16398 | 11864 |
| Avg. GPS Trajectory Length ($m$) | 2829.16 | 2797.26 |
| Avg. Route Trajectory Length | 15.26 | 15.96 |
| Avg. Road Travel Speed ($m/s$) | 6.91 | 6.22 |
| Avg. Trajectory Travel Time ($s$) | 426.31 | 467.47 |

### B.1.2 Compared Methods

We compare SMARTraj[2] against several baseline methods that employ self-supervised training approaches and are designed for general-purpose trajectory representation learning, suitable for multiple downstream tasks. These baselines offer a solid foundation for assessing the advantages of our approach under consistent experimental conditions.

- Random: it initializes trajectory representations randomly, providing a reference for understanding the performance improvements achieved by more sophisticated models.

- Word2Vec [26]: it learns representations using the skip-gram model, which captures semantic similarities between road segments by treating them as words in a sequence, based on co-occurrence statistics within trajectories.

- Node2Vec [15]: it learns node representations in a graph via biased random walks, which explore node neighborhoods to capture both local and global structural properties.

- Transformer [39]: it employs a self-attention mechanism to model complex dependencies in sequential data.

---

[3]http://geodata.pku.edu.cn

- BERT [10]: it is pre-trained to learn deep bidirectional representations by conditioning on both left and right contexts at all layers.
- Toast [7]: it first pre-trains node embeddings using Node2Vec, then fine-tunes the representations with Transformer, incorporating auxiliary traffic context information.
- JCRLNT [25]: it employs separate graph and trajectory encoders, training the model on three comparative tasks to refine the quality of the learned representations.
- START [19]: it introduces a trajectory encoder that incorporates travel semantics and temporal continuity, trained with two self-supervised tasks to improve trajectory representation quality.
- JGRM [24]: it combines GPS trace data with route traces to model road network constraints, capturing both spatial and temporal dynamics.
- MVTraj [33]: it captures multiple structural and semantic aspects of trajectory data from three different spatial views, offering a rich and diverse representation suited for various downstream tasks.

None of the baseline methods effectively tackle the challenge of multi-city structural heterogeneity, a critical factor for generalization across urban scenes. Consequently, these methods fail to leverage datasets that encompass multiple cities, and are instead trained on data from a single city per experiment.

### B.1.3 Detail of Evaluation Metrics

We employ a range of evaluation metrics to comprehensively compare the performance of different methods across various tasks. These metrics are designed to capture different aspects of prediction accuracy, from classification performance to regression error.

- Micro-F1: This metric aggregates the contributions of all classes into a single overall F1 value, providing a balance between precision and recall across the entire dataset. It is calculated by summing the true positives (TP), false positives (FP), and false negatives (FN) across all classes to derive the overall precision and recall.

$$Micro - F1 = \frac{2 \times Precision_{all} \times Recall_{all}}{Precision_{all} + Recall_{all}}$$

This metric is useful when the dataset contains a large class imbalance, as it treats all instances equally.

- Macro-F1: Unlike Micro-F1, Macro-F1 treats each class equally by calculating the F1 score for each class independently and then averaging these scores:

$$Macro - F1 = \frac{1}{N} \sum_{i=1}^{N} F1_i$$

where $F1_i$ is the F1 score for the $i$-th class, and $N$ is the total number of classes. Macro-F1 is particularly effective when dealing with imbalanced datasets, as it ensures that all classes are equally represented in the final metric.

- Mean Absolute Error (MAE): MAE quantifies the average magnitude of the errors in a set of predictions, providing a straightforward interpretation of the average error magnitude:

$$MAE = \frac{1}{n} \sum_{i=1}^{n} |y_i - \hat{y}_i|$$

where $y_i$ is the true value, $\hat{y}_i$ is the predicted value, and $n$ is the total number of observations. MAE is sensitive to small deviations and provides a clear measure of average prediction accuracy.

- Root Mean Squared Error (RMSE): RMSE measures the square root of the average squared differences between predicted and true values, emphasizing larger errors due to the squaring of differences:

$$RMSE = \sqrt{\frac{1}{n} \sum_{i=1}^{n} (y_i - \hat{y}_i)^2}$$

This metric is useful for capturing the variance in prediction errors, with larger errors being penalized more than smaller ones.

- Accuracy@k ($Acc@k$): This metric evaluates whether the true label appears in the top-$k$ predicted labels for each instance:

$$Acc@k = \frac{1}{n} \sum_{i=1}^{n} \mathbb{I}(y_i \in \hat{y}_i^{(k)})$$

where $y_i$ is the true label, $\hat{y}_i^{(k)}$ represents the set of top-$k$ predicted labels, and $\mathbb{I}(\cdot)$ is an indicator function returning 1 if the true label $y_i$ is within the top-$k$ predictions, and 0 otherwise. $Acc@k$ is particularly relevant in ranking and recommendation tasks, offering insight into the effectiveness of the model in providing accurate top-$k$ suggestions.

### B.1.4 Implementation Details

Our evaluation follows a two-stage process to ensure robustness and fairness. In the first stage, the encoder is pre-trained on a large set of unlabeled trajectory data (e.g., 50K trajectories from the Chengdu dataset) to learn informative trajectory representations. In the second stage, a smaller labeled subset (e.g., 12K labeled trajectories from the Chengdu dataset) is used to fine-tune the model and train task-specific models for classification or regression. These task-specific models predict outputs such as road labels, travel time, destination road segment IDs, or destination grid indices.

To optimize the model, we use the AdamW optimizer for both pre-training and fine-tuning. The training process spans 70 epochs with a batch size of 64. The initial learning rate is set to 0.0001, and we adopt a warm-up policy that linearly increases the learning rate during the first five epochs. Afterward, a cosine annealing schedule is employed to gradually reduce the learning rate in the subsequent epochs.

To enhance the model's robustness, we introduce a masking mechanism during training. Specifically, we apply a masking length of 2 with a probability of 20%.

### B.2 Additional Experiments

### B.2.1 Additional Performance Comparison

Table 4: Overall Performance in Chengdu .

| Method | Road Label Micro-F1 / Macro-F1 | Travel Time MAE / RMSE | Destination Road Acc@1 / Acc@5 | Destination Grid Acc@1 / Acc@5 |
|---|---|---|---|---|
| Random | 0.4363 / 0.3152 | 112.3310 / 141.6182 | 0.651 / 0.7795 | ✗ / ✗ |
| Word2vec | 0.5857 / 0.5767 | 85.4754 / 113.8926 | 0.6093 / 0.7717 | ✗ / ✗ |
| Node2vec | 0.5535 / 0.5306 | 85.9276 / 114.4905 | 0.604 / 0.7611 | ✗ / ✗ |
| Transformer | 0.3753 / 0.3460 | 88.3027 / 117.2306 | 0.6297 / 0.7969 | ✗ / ✗ |
| BERT | 0.5516 / 0.5363 | 86.8267 / 115.4532 | 0.5994 / 0.7755 | ✗ / ✗ |
| Toast | 0.7145 / 0.6755 | 92.2311 / 125.6123 | 0.5966 / 0.773 | ✗ / ✗ |
| JCLRNT | 0.6100 / 0.6037 | 90.9430 / 116.6238 | 0.5147 / 0.7953 | ✗ / ✗ |
| START | 0.3526 / 0.1869 | 112.0348 / 148.3855 | 0.6872 / 0.7764 | ✗ / ✗ |
| JGRM | 0.7198‡ / 0.7228‡ | 82.8468‡ / 110.3405‡ | 0.7304† / 0.873† | ✗ / ✗ |
| MVTraj | 0.7206† / 0.7326† | 48.5581† / 71.8248† | 0.7021‡ / 0.8597‡ | 0.7927† / 0.9105† |
| SMARTraj[2] | **0.7461 / 0.7473** | **34.3311 / 54.7641** | **0.7395 / 0.881** | **0.8082 / 0.9289** |

\* **Bold** denotes the best result, † and ‡ denotes the second and third best result.

Tab. 4 presents the ablation study results on the Chengdu dataset, which are consistent with those obtained in Xi'an. The results demonstrate that each proposed component contributes significantly to overall model performance. Moreover, the Chengdu results further validate the robustness and stability of SMARTraj[2] under varying urban conditions and data distributions.

### B.2.2 Additional Model Analysis

Fig. 5 presents the model analysis results on the Xi'an dataset, which are consistent with those observed in Chengdu. The results confirm that pre-training substantially enhances model performance.

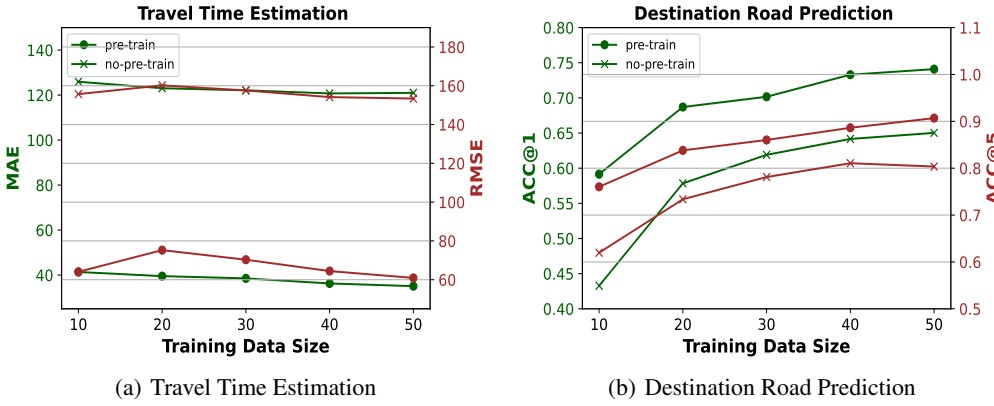

| (a) Travel Time Estimation | (b) Destination Road Prediction |

Figure 5: Effect of Pre-training in Xi'an.

Compared to training from scratch, the pre-trained model exhibits faster convergence and lower prediction errors, highlighting the effectiveness of self-supervised trajectory representation learning. Moreover, the performance gain remains stable even when the amount of labeled data is reduced, demonstrating the model's robustness and data efficiency in low-supervision scenarios.

### B.2.3 Additional Ablation Study

Table 5: Ablation Study in Chengdu.

| Method | Road Label Micro-F1 / Macro-F1 | Travel Time MAE / RMSE | Destination Road Acc@1 / Acc@5 | Destination Grid Acc@1 / Acc@5 |
|---|---|---|---|---|
| SMARTraj[2] | 0.7461 / 0.7473 | 34.3311 / 54.7641 | 0.7395 / 0.8810 | 0.8082 / 0.9289 |
| w/o diff loss | 0.7320 / 0.7314 | 35.0011 / 57.7368 | 0.6006 / 0.7817 | 0.6777 / 0.8558 |
| w/o gating | 0.7378 / 0.7399 | 34.4141 / 55.7714 | 0.6037 / 0.7868 | 0.6796 / 0.8583 |

Tab. 5 presents the ablation study results on the Chengdu dataset, which are consistent with those obtained in Xi'an. The results clearly demonstrate that both the difference loss ($\mathcal{L}_{diff}$) and the personalized gating mechanism play essential roles in the model's performance. These findings collectively validate the effectiveness and complementarity of both modules in enhancing the model's generalization capability.

We have also conducted additional ablation experiments where the model is trained using only data from a single city, and compared it with the multi-city training setting. The results for Xi'an and Chengdu are presented in Tab. 6.

Across all tasks and evaluation metrics, multi-city learning consistently outperforms single-city training. This demonstrates that incorporating data from multiple cities enables the model to learn more generalized and transferable patterns, leading to better performance even on individual city tasks. These findings validate the effectiveness and necessity of multi-city learning.

Table 6: Ablation Study.

| City | Method | Road Label Micro-F1 / Macro-F1 | Travel Time MAE / RMSE | Destination Road Acc@1 / Acc@5 | Destination Grid Acc@1 / Acc@5 |
|---|---|---|---|---|---|
| Xi'an | multi-city | 0.8407 / 0.8298 | 35.0689 / 60.9156 | 0.7409 / 0.9069 | 0.6675 / 0.8392 |
| | single-city | 0.8325 / 0.8144 | 47.7116 / 76.4029 | 0.6973 / 0.7915 | 0.6569 / 0.8133 |
| Chengdu | multi-city | 0.7461 / 0.7473 | 34.3311 / 54.7641 | 0.7395 / 0.8810 | 0.8082 / 0.9289 |
| | single-city | 0.7243 / 0.7273 | 43.6200 / 65.1772 | 0.7011 / 0.8610 | 0.7887 / 0.8991 |

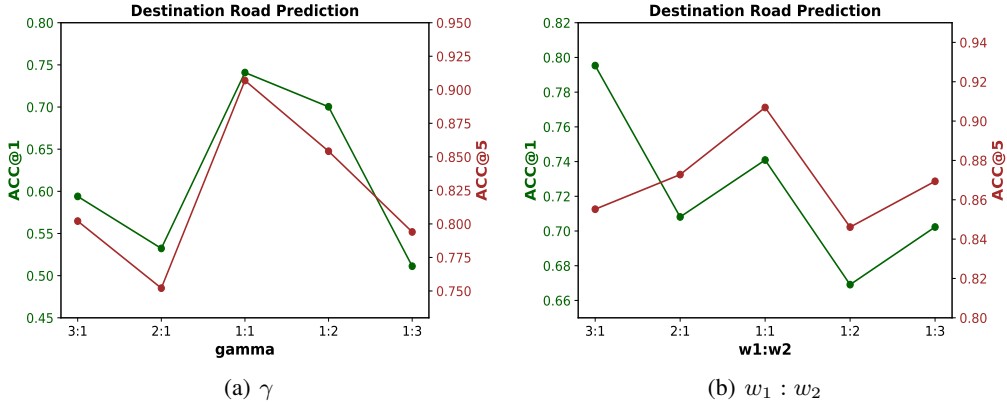

(a) $\gamma$                    (b) $w_1 : w_2$

Figure 6: Parameter Sensitivity Analysis on Destination Road Prediction in Xi'an.

### B.2.4  Additional Parameter Sensitivity

Fig. 6 presents the sensitivity analysis results for the destination road prediction task in Xi'an, which are consistent with those obtained for travel time estimation. These consistent trends across tasks further demonstrate the stability and robustness of the proposed model with respect to hyperparameter choices.

### B.2.5  Model Efficiency

Table 7: Efficiency Comparison on Xi'an.

|  | Model Size (MBytes) | Train Time (min/epoch) | Inference Time (milliseconds) |
|---|---|---|---|
| Random | - | - | 0.374 |
| Word2Vec | 8.0 | 0.2 | 0.404 |
| Node2Vec | 7.6 | 0.2 | 0.328 |
| Transformer | 14 | 1.5 | 0.940 |
| BERT | 433 | 3.5 | 1.260 |
| Toast | 27 | 7.2 | 2.152 |
| JCLRNT | 13 | 1.3 | 0.792 |
| START | 94 | 17 | 4.700 |
| JGRM | 33 | 15 | 4.220 |
| MVTraj | 207 | 35 | 10.560 |
| SMARTraj[2] | 477 | 54 | 11.65 |

As shown in Tab. 7, while our method has a larger model size compared to some baselines, its inference time remains within a practical range. These results suggest that our approach is feasible for real-world deployment.

