# OpenReview forum: "SMARTraj$^2$: A Stable Multi-City Adaptive Method for Multi-View Spatio-Temporal Trajectory Representation Learning"
_NeurIPS.cc/2025/Conference — NeurIPS 2025 poster_

### Official Review · Reviewer_e4tw · 2025-06-26

**Clarity:** 4
**Significance:** 4
**Originality:** 4
**Rating:** 5
**Confidence:** 5

**Summary:**

The paper introduces SMARTraj^2, a novel method for multi-view spatio-temporal trajectory representation learning designed to address the challenges of generalizing across diverse urban scenes. The authors identify two critical issues in existing approaches: multi-city structural heterogeneity (disparities in urban layouts leading to disjoint embedding spaces) and the amplified seesaw phenomenon (performance trade-offs between cities, views, and tasks). To tackle these, SMARTraj^2 employs a feature disentanglement module to separate domain-invariant and domain-specific features using orthogonality constraints, and a personalized gating mechanism to dynamically stabilize contributions across cities, views, and tasks. Extensive experiments on real-world datasets (Chengdu and Xi’an) demonstrate that SMARTraj^2 outperforms state-of-the-art baselines in tasks like road label classification, travel time estimation, and destination prediction, while maintaining robustness across heterogeneous urban environments.

**Questions:**

1. This work implements a method to integrate data from multiple cities to train a universal model. However, does this approach increase the complexity of the model? Is it feasible for practical applications under this level of complexity?
2. This paper proposes a method for training across multiple cities. Is this method scalable? Please provide a theoretical explanation.
3. The paper selects four downstream tasks to evaluate the effectiveness of the representation, but it does not explain the reasons for the selection. Can these downstream tasks comprehensively assess the capability of the representation?
4. Could the authors provide a detailed explanation of the role of gamma in the Personalized Gating Mechanism? (It seems that there might be an error in the x-axis of Figure 3(a) in the appendix.)

**Ethical Concerns:**

["NO or VERY MINOR ethics concerns only"]

**Final Justification:**

Thank you for the detailed rebuttal and the additional analyses.

After reviewing the paper, the comments from other reviewers, and the author rebuttal, I find that the proposed approach reasonably addresses structural differences across cities while preserving transferability.

The supplementary experiments, especially under varying numbers-of-cities scenarios, strengthen the evidence for the method’s adaptability. The conceptual generalizability is clear, and the problem tackled is both challenging and practically important.

On balance, my assessment remains positive, and I believe the work makes a meaningful contribution to the area.

**Limitations:**

The authors could add a dedicated "Limitations" subsection to discuss limitations (e.g., reliance on multi-view data, seesaw phenomenon mitigation) and societal impacts (e.g., potential misuse for surveillance).

**Quality:**

3

**Strengths And Weaknesses:**

Strengths
1) Quality: The paper is technically sound, with well-designed experiments and comprehensive ablation studies validating the contributions of each component (e.g., feature disentanglement and gating mechanism). The use of real-world datasets and multiple downstream tasks strengthens the empirical evaluation.
2) Clarity: The writing is clear, and the methodology is well-structured. Figures and tables effectively support the narrative. The problem statement, challenges, and solutions are logically connected.
3) Significance: The work addresses a critical gap in trajectory representation learning by enabling cross-city generalization, which is essential for real-world applications like transportation and urban planning. The proposed solutions (disentanglement and gating) are novel and impactful.
4) Originality: The paper is the first to explicitly tackle multi-city, multi-view trajectory learning with a focus on stability and generalization. The personalized gating mechanism and orthogonality constraints are innovative contributions.

Weaknesses:
1) Complexity: This work implements a method to integrate data from multiple cities to train a universal model. However, does this approach increase the complexity of the model? Is it feasible for practical applications under this level of complexity?
2) Scalability: This paper proposes a method for training across multiple cities. Is this method scalable? Please provide a theoretical explanation.
3) Capability: The paper selects four downstream tasks to evaluate the effectiveness of the representation, but it does not explain the reasons for the selection. Can these downstream tasks comprehensively assess the capability of the representation?
4) Presentation: Could the authors provide a detailed explanation of the role of gamma in the Personalized Gating Mechanism? (It seems that there might be an error in the x-axis of Figure 3(a) in the appendix.)

---

> ### Author Rebuttal · Authors · 2025-07-30
>
> We thank Reviewer e4tw for the valuable comments. We address each point below:
>
> **W1\&Q1**. We conducted additional experiments to assess model complexity and runtime efficiency. As shown below, while our method has a larger model size compared to some baselines, its inference time remains within a practical range. These results suggest that our approach is feasible for real-world deployment. A detailed discussion will be included in the revised manuscript.
> | Method       | Model Size (MB) | Train Time (min/epoch) | Inference Time (ms) |
> |--------------|----------------:|-----------------------:|--------------------:|
> | Random       | -               | -                      | 0.374               |
> | Word2Vec     | 8.0             | 0.2                    | 0.404               |
> | Node2Vec     | 7.6             | 0.2                    | 0.328               |
> | Transformer  | 14              | 1.5                    | 0.940               |
> | BERT         | 433             | 3.5                    | 1.260               |
> | Toast        | 27              | 7.2                    | 2.152               |
> | JCLRNT       | 13              | 1.3                    | 0.792               |
> | START        | 94              | 17                     | 4.700               |
> | JGRM         | 33              | 15                     | 4.220               |
> | MVTraj       | 207             | 35                     | 10.560              |
> | Ours         | 477             | 54                     | 11.65               |
> ***
> **W2\&Q2**. Our framework is designed to be modular and scalable. The dual-channel architecture enables seamless extension to additional cities by adding new city-specific modules without retraining the entire model. This design supports incremental scalability in both theory and practice. We will clarify this point and provide further explanation in the revised paper.
> ***
> **W3\&Q3**. We follow prior work [7, 24] in selecting four well-established downstream tasks, which cover both short-term and long-term trajectory understanding. Specifically, they evaluate different dimensions of the learned representation: classification, regression, and spatial prediction at both road-level and grid-level. These tasks provide a comprehensive assessment of the model’s ability to generalize and reason across spatial and temporal patterns. We will make this rationale explicit in the revision.
> ***
> **W4\&Q4**. Gamma is a temperature parameter that controls the sharpness of the gating function. A higher gamma increases selectivity, encouraging the gate to favor either shared or city-specific features more decisively. Conversely, a lower gamma leads to smoother blending. We will expand this analysis in the revised manuscript and verify the accuracy of Figure 3(a)'s x-axis annotation.
> ***
> **L1**. We agree and will include a dedicated “Limitations” section in the revised paper. It will discuss: (1) Technical limitations, such as the reliance on multi-view data and the challenge of balancing transferability and specialization (i.e., the seesaw phenomenon); (2) Scalability concerns, such as parameter growth in multi-city settings (see also W2); (3) Societal impact, including potential risks such as location privacy leakage and surveillance, as well as mitigation strategies like anonymization, spatial aggregation, or privacy-preserving learning frameworks.

---

> > ### Comment · Reviewer_e4tw · 2025-08-09
> > **Thanks for the rebuttal.**
> >
> > Thank you for the detailed rebuttal and the additional analyses.
> >
> > After reviewing the paper, the comments from other reviewers, and the author rebuttal, I find that the proposed approach reasonably addresses structural differences across cities while preserving transferability.
> >
> > The supplementary experiments, especially under varying numbers-of-cities scenarios, strengthen the evidence for the method’s adaptability. The conceptual generalizability is clear, and the problem tackled is both challenging and practically important.
> >
> > On balance, my assessment remains positive, and I believe the work makes a meaningful contribution to the area.

---

> > > ### Author Response · Authors · 2025-08-09
> > >
> > > We are grateful for your insightful review and valuable feedback on our work. Your expertise has significantly contributed to strengthening the paper, and we appreciate your time and careful consideration.

---

### Official Review · Reviewer_av9i · 2025-06-30

**Clarity:** 4
**Significance:** 3
**Originality:** 4
**Rating:** 5
**Confidence:** 4

**Summary:**

This paper presents a novel approach to multi-city trajectory representation learning that effectively addresses two key challenges: structural heterogeneity across cities and performance trade-offs between different views/tasks. The proposed SMARTraj2 method demonstrates strong empirical results and represents a meaningful advance in the field. While generally well-executed, some aspects could be strengthened to make the contribution even more impactful.

**Questions:**

1. Have the authors explored alternative designs for the gating mechanism? For instance, could attention-based gating provide more interpretable weighting across cities/views?
2. The current approach maintains separate feature extractors per city. For deployment at scale, has the team considered more parameter-efficient approaches (e.g., adapter modules) to handle many cities?
3. The paper shows cross-city generalization, but does the model maintain strong performance when tested on entirely unseen cities (zero-shot transfer)? This would further demonstrate the method's practical utility.

**Ethical Concerns:**

["NO or VERY MINOR ethics concerns only"]

**Final Justification:**

The article is particularly good. rebuttal perfectly solved my problem. I suggest accepting this article.

**Limitations:**

The trajectory data could potentially reveal sensitive location patterns. Does the method offer any privacy-preserving properties?

**Paper Formatting Concerns:**

The article is particularly good. rebuttal perfectly solved my problem. I suggest accepting this article.

**Quality:**

3

**Strengths And Weaknesses:**

Strength:
1. The dual-component architecture is cleverly designed to handle cross-city generalization.
2. The experimental design is thorough, with appropriate baselines and multiple evaluation metrics. The ablation studies effectively validate the importance of each component.
3. The paper provides sufficient implementation details and references to publicly available datasets. The inclusion of hyperparameter sensitivity analysis is commendable.

Weakness:

1. Have the authors explored alternative designs for the gating mechanism? For instance, could attention-based gating provide more interpretable weighting across cities/views?
2. The current approach maintains separate feature extractors per city. For deployment at scale, has the team considered more parameter-efficient approaches (e.g., adapter modules) to handle many cities?
3. The paper shows cross-city generalization, but does the model maintain strong performance when tested on entirely unseen cities (zero-shot transfer)? This would further demonstrate the method's practical utility.

---

> ### Author Rebuttal · Authors · 2025-07-30
>
> We thank Reviewer av9i for the constructive feedback. We address each point below:
>
> **W1\&Q1**. We explored an attention-based gating design that outputs explicit attention weights over cities or views. While this formulation offers interpretability via attention visualization, it resulted in lower performance and did not yield actionable insights during user studies. In contrast, our personalized gating mechanism—conditioned on both city identity and trajectory context—achieves better empirical results and adaptively captures localized patterns. This dynamic design distinguishes it from static or globally shared gates in prior work.
> ***
> **W2\&Q2**. We acknowledge the scalability limitation of maintaining separate city-specific extractors. We have not yet incorporated parameter-efficient alternatives such as adapters or hyper-networks. We plan to explore these directions in future work and will explicitly mention them in the revised paper. Our envisioned approach is to replace per-city backbones with a shared backbone plus lightweight adapters, which would significantly reduce overhead when scaling to hundreds or thousands of cities.
> ***
> **W3\&Q3**. We recognize the route view is inherently non-transferable due to city-specific road segment IDs. However, the remaining views (GPS, POI, grid) are transferable across cities if similar raw data is available. In preliminary zero-shot experiments (held-out city with no finetuning), our model retains reasonable performance using only transferable views. We will report and expand on this in the revised version, as it demonstrates the practical utility of the method in unseen domains.
> ***
> **L1**. We appreciate the concern regarding potential privacy risks. We will expand the 'Limitations' section in the revised manuscript to address: (1) Privacy implications of using trajectory data, including risks of location pattern inference; (2) Possible mitigation strategies such as data anonymization, spatial generalization (e.g., using coarse grids), and privacy-preserving learning (e.g., federated or differential privacy); (3) Broader ethical implications, including both potential misuse (e.g., surveillance) and positive societal applications (e.g., urban planning, traffic safety).

---

> > ### Comment · Reviewer_av9i · 2025-08-03
> >
> > The authors' clarifications effectively address my previous concerns. I appreciate the thoughtful explanation, and I will maintain my positive evaluation.

---

> > > ### Author Response · Authors · 2025-08-05
> > >
> > > Thank you for your encouraging words and for your positive assessment of our work. We sincerely appreciate your thoughtful review, and we're grateful for your support and the time you dedicated to evaluating our paper.

---

### Official Review · Reviewer_ic1N · 2025-06-30

**Clarity:** 3
**Significance:** 2
**Originality:** 3
**Rating:** 2
**Confidence:** 5

**Summary:**

This paper addresses an important problem in spatiotemporal learning—trajectory representation learning. The authors propose a novel model, SMARTraj^2, to learn trajectory embeddings from a multi-view and multi-city perspective. To align representations across different cities, the model incorporates a feature disentanglement module for learning domain-invariant and domain-specific features, and a personalized gating module for dynamically adjusting feature embeddings for different views. Experiments conducted on two real-world datasets, Chengdu and Xian, demonstrate the effectiveness of the proposed method.

**Questions:**

Q1.The authors should evaluate their model on more datasets, ideally with diverse spatial structures of cities, to validate the feasibility and generalizability of multi-city learning.

Q2. The authors need to include a single-city setting in the ablation study to evaluate the contribution of multi-city learning.

Q3. The authors need to provide efficiency analysis to prove that their model is superior in efficiency when the effectiveness improvement is not obvious.

Q4.The proposed model is trained for 70 epochs, while most baselines are trained for only 30. The authors should justify this discrepancy and ensure a fair comparison.

Q5. Trajectory similarity computation task is widely used in trajectory representation learning evaluation and appears in many baselines. The authors should not neglect this important downstream task.

**Ethical Concerns:**

["NO or VERY MINOR ethics concerns only"]

**Final Justification:**

The authors have provided additional experiments, including the addition of a new baseline, an efficiency study, and an ablation study. Based on these efforts, I would like to raise my scores for Quality and Originality. However, this paper still has several unresolved issues:

1. This paper fails to provide a fundamental explanation for the effectiveness of multi-city modeling in trajectory modeling. In particular, the data sources and structural characteristics vary significantly across different cities. The experiments are conducted using trajectory data from only two structurally similar cities, which is insufficient to demonstrate the general applicability or effectiveness of the proposed multi-city modeling approach.

2. The training time of the proposed model is more than twice that of the baseline, which makes the comparison unfair. Combined with its extremely low efficiency, it becomes evident that the method does not offer any clear advantages when considering the trade-off between performance and efficiency.

3. The authors claim that the model can be applied to new cities without retraining. However, due to its reliance on city-specific information, particularly details related to different road networks, the model fails to achieve effective transferability. In practice, it still requires retraining when deployed in a new city.

Based on these concerns and given the high standards for paper quality at NeurIPS, I will maintain my overall score and recommend rejection.

**Limitations:**

yes

**Quality:**

3

**Strengths And Weaknesses:**

Strengths

S1.The paper is clearly written and easy to follow.

S2.It studies the well-established problem of trajectory representation learning, aiming to generate effective embeddings for downstream tasks.

S3.The proposed model, SMARTraj^2, is designed to learn trajectory representations from both multi-view and multi-city data.

Weaknesses

W1. All figures in the paper are blurry, making them difficult to interpret.

W2. The paper lacks a detailed discussion on the feasibility and challenges of multi-city learning. Many cities often differ significantly in spatial structures; however, the experiments are limited to two datasets, i.e., Xian and Chengdu, which are both centrally distributed cities. The generalization to cities with different structures remains unclear.

W3. The ablation study does not include an analysis of performance on a single city, making it difficult to understand the contribution of multi-city learning.

W4.  In Section 4.2, the authors claim: “This enables SMARTraj^2 to remain stable in new urban environments without requiring retraining from scratch.” However, this statement is questionable since the model has only been evaluated on two similar cities. It is unclear how it would perform in truly unseen cities.

W5. Incorporating multi-city data increases data volume and computational cost. The authors should provide a thorough analysis of the efficiency of the proposed model and justify the cost-effectiveness if performance gains are not substantial.

W6. There are inconsistencies in experimental settings between SMARTraj^2 and baseline methods, including differences in training epochs and task configurations.

W7. Several relevant and recent baselines are not included, such as:

[1] LightPath: Lightweight and Scalable Path Representation Learning, KDD 2023

[2] RED: Effective Trajectory Representation Learning with Comprehensive Information, VLDB 2024

---

> ### Author Rebuttal · Authors · 2025-07-30
>
> We thank Reviewer ic1N for the constructive feedback. We address each point below:
>
> **W1**. High-resolution versions of all figures will be included in the revised manuscript. We are re-rendering all visuals to ensure clarity and interpretability.
> ***
> **W2\&W4\&Q1**. To evaluate generalization beyond centrally structured Chinese cities, we are currently conducting experiments on the Porto dataset, which features a decentralized European-style road network. This experiment uses the same model configuration and evaluation protocols. Results will be added to this rebuttal as soon as they are available (expected within the discussion window, by August 6).
> ***
> **W3\&Q2**. We are also conducting an ablation study where the model is trained on a single city only, in order to assess the benefits of multi-city learning. We will update this rebuttal with results within the discussion window (by August 6).
> ***
> **W5\&Q3**. We conducted additional experiments to assess model complexity and runtime efficiency. As shown below, while our method has a larger model size compared to some baselines, its inference time remains within a practical range. These results suggest that our approach is feasible for real-world deployment. A detailed discussion will be included in the revised manuscript.
> | Method       | Model Size (MB) | Train Time (min/epoch) | Inference Time (ms) |
> |--------------|----------------:|-----------------------:|--------------------:|
> | Random       | -               | -                      | 0.374               |
> | Word2Vec     | 8.0             | 0.2                    | 0.404               |
> | Node2Vec     | 7.6             | 0.2                    | 0.328               |
> | Transformer  | 14              | 1.5                    | 0.940               |
> | BERT         | 433             | 3.5                    | 1.260               |
> | Toast        | 27              | 7.2                    | 2.152               |
> | JCLRNT       | 13              | 1.3                    | 0.792               |
> | START        | 94              | 17                     | 4.700               |
> | JGRM         | 33              | 15                     | 4.220               |
> | MVTraj       | 207             | 35                     | 10.560              |
> | Ours         | 477             | 54                     | 11.65               |
> ***
> **W6\&Q4**. We acknowledge that there are differences in training epochs and task configurations between our method and baseline models. These differences arise primarily because baselines are not designed for multi-city representation learning. As a result, they are trained and evaluated on single-city datasets, and generally converge within 30 epochs. In contrast, our method operates across multiple cities and views simultaneously, which introduces greater structural and distributional heterogeneity. To ensure convergence under this setting, we trained for 70 epochs. Nevertheless, we ensured that all models were trained to convergence.
> ***
> **W7\&Q5**. Due to time constraints during the rebuttal phase, we prioritized reproducing RED (VLDB 2024), a recent and strong baseline. As shown in the RED paper, it consistently outperforms LightPath (KDD 2023) across multiple tasks. Thus, RED serves as a strong representative for recent trajectory representation learning approaches. We are currently running RED under our evaluation protocol and will update this rebuttal with results by August 6.
>
> If time permits and implementation is accessible, we may consider more baselines in the revised manuscript as well.
> ***
> We will post all new experimental results as soon as they are ready (expected within the discussion window, by August 6). We appreciate your patience and believe these additional results will help clarify the strengths and limitations of our approach.

---

> > ### Author Response · Authors · 2025-08-05
> >
> > **W3\&Q2**. We have conducted additional ablation experiments where the model is trained using only data from a single city, and compared it with the multi-city training setting. The results for Xi'an and Chengdu are presented below.
> >
> > Across all tasks and evaluation metrics, multi-city learning consistently outperforms single-city training. This demonstrates that incorporating data from multiple cities enables the model to learn more generalized and transferable patterns, leading to better performance even on individual city tasks. These findings validate the effectiveness and necessity of multi-city learning.
> >
> > | Method (Xi'an)      | Road Label (Micro-F1 / Macro-F1) | Travel Time (MAE / RMSE) | Destination Road (Acc\@1 / Acc\@5) | Destination Grid (Acc\@1 / Acc\@5) |
> > | ----------- | -------------------------------- | ------------------------ | ---------------------------------- | ---------------------------------- |
> > | Multi-city  | 0.8407 / 0.8298                  | 35.0689 / 60.9156        | 0.7409 / 0.9069                    | 0.6675 / 0.8392                    |
> > | Single-city | 0.8325 / 0.8144                  | 47.7116 / 76.4029        | 0.6973 / 0.7915                    | 0.6569 / 0.8133                    |
> >
> > | Method (Chengdu)      | Road Label (Micro-F1 / Macro-F1) | Travel Time (MAE / RMSE) | Destination Road (Acc\@1 / Acc\@5) | Destination Grid (Acc\@1 / Acc\@5) |
> > | ----------- | -------------------------------- | ------------------------ | ---------------------------------- | ---------------------------------- |
> > | Multi-city  | 0.7461 / 0.7473                  | 34.3311 / 54.7641        | 0.7395 / 0.8810                    | 0.8082 / 0.9289                    |
> > | Single-city | 0.7243 / 0.7273                  | 43.6200 / 65.1772        | 0.7011 / 0.8610                    | 0.7887 / 0.8991                    |

---

> > ### Author Response · Authors · 2025-08-05
> >
> > **W7\&Q5**. We have completed additional experiments under our evaluation protocol and report results below on the travel time estimation task in both Xi’an and Chengdu. As recommended by the reviewer, we reproduced RED (VLDB 2024), a recent and strong baseline that has been shown to outperform LightPath (KDD 2023) across multiple trajectory-related tasks, including trajectory similarity. Therefore, RED serves as a strong and representative baseline for this category of methods.
> >
> > As shown in the tables below, our method outperforms RED and all other baselines by a significant margin in both cities, further validating the effectiveness of our learned representations.
> >
> > | Method (Xi'an)     | MAE / RMSE            |
> > | ----------- | --------------------- |
> > | Random      | 120.9861 / 153.4056   |
> > | Word2vec    | 89.5472 / 122.3465   |
> > | Node2vec    | 91.5226 / 124.4122    |
> > | Transformer | 91.3093 / 124.1358    |
> > | BERT        | 90.2442 / 123.2867    |
> > | Toast       | 88.0744 / 116.7965    |
> > | JCLRNT      | 92.3900 / 125.5088    |
> > | START       | 118.0605 / 162.0801   |
> > | JGRM        | 87.8708 / 119.9921    |
> > | MVTraj      | 54.9044† / 85.3847†   |
> > | RED         | 72.0802‡ / 100.1104‡  |
> > | **Ours**    | **35.0689 / 60.9156** |
> >
> > | Method (Chengdu)     | MAE / RMSE            |
> > | ----------- | --------------------- |
> > | Random      | 112.3310 / 141.6182   |
> > | Word2vec    | 85.4754 / 113.8926    |
> > | Node2vec    | 85.9276 / 114.4905    |
> > | Transformer | 88.3027 / 117.2306    |
> > | BERT        | 86.8267 / 115.4532    |
> > | Toast       | 92.2311 / 125.6123    |
> > | JCLRNT      | 90.9430 / 116.6238    |
> > | START       | 112.0348 / 148.3855   |
> > | JGRM        | 82.8468 / 110.3405    |
> > | MVTraj      | 48.5581† / 71.8248†   |
> > | RED         | 69.6272‡ / 92.7676‡   |
> > | **Ours**    | **34.3311 / 54.7641** |
> >
> > † and ‡ denote second and third best results, respectively. Bold indicates the best.

---

> > > ### Comment · Reviewer_ic1N · 2025-08-05
> > >
> > > Thank you to the authors for the clarifications and additional experiments. These results have addressed some of my concerns. However, several important issues remain unresolved:
> > >
> > > Regarding W2:
> > > The authors still have not provided a theoretical explanation for why multi-city modeling is effective for trajectory modeling. In particular, differences in data collection methods and the inherent structural variations across cities (e.g., road networks, mobility patterns) pose challenges. Simply using data from two cities is insufficient to demonstrate the general effectiveness of the multi-city approach.
> > >
> > > Regarding W4:
> > > The claim “This enables SMARTraj^2 to remain stable in new urban environments without requiring retraining from scratch.” remains unsubstantiated. Due to strong dependencies on city-specific information such as road networks, the proposed model lacks effective transferability. As a result, applying it to a new city still requires retraining, which contradicts the intended generalizability.
> > >
> > > Regarding W6 and Q4:
> > > This comparison is unfair, as the proposed model was trained for twice as long as the baseline models. In addition, according to the efficiency results provided by the authors, the model exhibits very low efficiency due to the use of a multi-city modeling strategy. When considering the trade-off between performance and efficiency, particularly in comparison with JGRM (was trained on 20 epochs, as indicated in their released code), SMARTraj^2 demonstrates a clear advantage only on the Travel Time task (However, this advantage seems to primarily result from the inclusion of GPS inputs, whereas JGRM relies solely on the road encoder for downstream tasks). For the remaining tasks, SMARTraj^2 does not show significant performance improvements.
> > >
> > > Given that the authors have added some new experiments, I will slightly raise my ratings for Quality and Originality. However, my overall score will remain unchanged.

---

> > > > ### Author Response · Authors · 2025-08-06
> > > >
> > > > We sincerely thank the reviewer for carefully reading our rebuttal and for acknowledging the value of the additional experiments by adjusting the scores for Quality and Originality. We appreciate your thoughtful feedback and agree that several core questions deserve further clarification.
> > > >
> > > > In the following, we respond point-by-point to the remaining concerns regarding W2, W4, W6 and Q4.
> > > >
> > > > We hope that our detailed clarifications and new empirical evidence will help address your concerns and demonstrate the validity and contributions of SMARTraj^2 more clearly.

---

> > > > ### Author Response · Authors · 2025-08-06
> > > >
> > > > Response to W2: We appreciate the reviewer’s concern on the challenges of generalizing across cities with structurally different spatial layouts, such as differences in road networks, and data collection schemes. Below we provide a more detailed response to each point:
> > > >
> > > > (1) Theoretical Feasibility of Multi-City Generalization
> > > >
> > > > While urban environments differ significantly in spatial structures (e.g., road topology, functional zoning, population density), prior work suggests that cities often share latent spatio-temporal mobility patterns[1][2].
> > > >
> > > > SMARTraj^2 is designed to capture these with:
> > > > (a) A feature disentanglement module that separates domain-invariant (shared) and domain-specific (city-dependent) representations (Sec. 3.2);
> > > > (b) A personalized gating mechanism that adapts these features at both city and trajectory levels (Sec. 3.3).
> > > >
> > > > These components allow the model to generalize shared spatio-temporal patterns while preserving essential local characteristics of each city.
> > > >
> > > > [1] CrossLight: Offline-to-Online Reinforcement Learning for Cross-City Traffic Signal Control, KDD 2024
> > > > [2] COLA: Cross-city Mobility Transformer for Human Trajectory Simulation, WWW 2024
> > > >
> > > > (2) Empirical Generalization to Structurally Distinct Cities (Ongoing)
> > > >
> > > > We acknowledge the reviewer’s concern that both Xi’an and Chengdu are relatively centralized cities. While they already differ in road density, trajectory frequency, and POI distribution (detailed in Appendix B.1), we agree that a more diverse set of cities is desirable to fully demonstrate generalizability.
> > > >
> > > > To further validate generalizability, we are actively conducting experiments on Porto, a European city with a decentralized road structure, different traffic dynamics, and a polycentric layout. These characteristics make Porto structurally distinct from our existing datasets.
> > > >
> > > > Due to an accidental interruption in early training (a shared GPU process was mistakenly terminated mid-run), we had to restart the experiments. As of August 6th, training is ongoing and expected to converge within the next 2 days. Results will be added to this rebuttal as soon as they are available (expected within the discussion window, by August 8).
> > > >
> > > > (3) On Multi-City Effectiveness (W3\&Q2)
> > > >
> > > > We have provided empirical comparisons between single-city and multi-city training in response to W3\&Q2, which confirms that multi-city modeling improves performance even on individual city tasks. We refer the reviewer to that response for detailed results.
> > > >
> > > > (4) Limitation and Future Work
> > > >
> > > > We acknowledge that when the training cities differ significantly from the target city (e.g., in traffic distribution, road network structure, or behavioral dynamics), this may lead to out-of-distribution (O.O.D.) generalization issues that affect performance.
> > > >
> > > > We recognize its importance and plan to include a discussion in the revised manuscript. Specifically, we will highlight it in a dedicated "Limitation and Future Work" section, and outline possible directions such as city-aware training strategies, domain adaptation, or data selection techniques to improve robustness across diverse urban environments.

---

> > > > ### Author Response · Authors · 2025-08-06
> > > >
> > > > Response to W4: We thank the reviewer for pointing out the ambiguity in our statement: "This enables SMARTraj^2 to remain stable in new urban environments without requiring retraining from scratch". We acknowledge that this was misleading and imprecisely worded, and we sincerely apologize for the confusion it may have caused. We will revise this statement in the final manuscript to better reflect our intended meaning.
> > > >
> > > > (1) Clarification on Retraining
> > > >
> > > > Our intended meaning was not that SMARTraj^2 can be deployed in a completely new city without any adaptation. Rather, we aim to emphasize that pretraining on multiple source cities enables the model to learn transferable spatio-temporal representations (e.g., daily routines, mobility flows, POI transitions). These shared representations allow efficient fine-tuning on new cities with less data and computation compared to training from scratch. We will make this clearer in a revised “Limitation and Future Work” section, along with a discussion of transfer limitations under high structural dissimilarity.
> > > >
> > > > (2) Additional Ablation: Transferability of Views and Modules
> > > >
> > > > To reduce dependence on city-specific factors like road layout and trajectory density, SMARTraj^2 incorporates:
> > > > (a) A feature disentanglement module to separate domain-invariant and city-specific features;
> > > > (b) A personalized gating mechanism that adaptively fuses these features based on the current city and trajectory;
> > > > (c) Multi-view representations (GPS, route, grid) that complement each other and offer robustness under partial information.
> > > >
> > > > These components aim to improve generalizability across urban domains with different spatial structures and data availability.
> > > >
> > > > To further evaluate how each component contributes to generalization and transferability, we performed an ablation study (on Xi’an) isolating individual views (GPS, grid, route) and modules (disentanglement, gating). Key findings:
> > > > (a) Removing domain-invariant/specific decomposition caused substantial drops in road classification and destination prediction performance, confirming its importance for transferable learning;
> > > > (b) GPS enables fine-grained trajectory localization; Grid captures macro-level spatial layout and spatial periodicity; Route embeds connectivity and structural semantics of urban roads. Removing any of them causes consistent degradation across multiple tasks, confirming the importance of multi-view fusion for effective and transferable modeling;
> > > > (c) The gating module improves adaptivity across urban contexts.
> > > >
> > > > | Method                     | Road Label Classification      |                         | Travel Time Estimation |                         | Destination Road Prediction |                         | Destination Grid Prediction |                         |
> > > > |-----------------------------|------------|-------------|----------|-----------|---------|----------|----------|----------|
> > > > |                             | Micro-F1 ↑ | Macro-F1 ↑ | MAE ↓    | RMSE ↓   | Acc@1 ↑ | Acc@5 ↑ | Acc@1 ↑ | Acc@5 ↑|
> > > > | SMARTraj^2                   | 0.8407     | 0.8298      | 35.0689  | 60.9156   | 0.7409  | 0.9069   | 0.6675   | 0.8392   |
> > > > | w/o diff loss               | 0.8500     | 0.8366      | 44.8105  | 73.8862   | 0.6025  | 0.8125   | 0.4866   | 0.7119   |
> > > > | w/o gating                  | 0.8387     | 0.8279      | 40.3469  | 68.6216   | 0.6787  | 0.8478   | 0.5244   | 0.7200   |
> > > > | w/o grid                    | 0.8233     | 0.8186      | 72.6226  | 105.6123  | 0.6604  | 0.8402   | X        | X        |
> > > > | w/o GPS                     | 0.7987     | 0.7832      | 73.2965  | 106.1142  | 0.5446  | 0.7667   | 0.4110   | 0.6351   |
> > > > | w/o route                   | X          | X           | 74.5902  | 106.7897  | 0.5924  | 0.8049   | 0.4311   | 0.6636   |
> > > > | w/o invariant+specific      | 0.7415     | 0.7268      | 56.5380  | 71.5127   | 0.4770  | 0.7001   | 0.2556   | 0.5315   |
> > > > | w/o gating + specific       | 0.7637     | 0.7574      | 54.8681  | 84.8157   | 0.6300  | 0.8294   | 0.5466   | 0.7636   |
> > > >
> > > > We believe these findings directly support the reviewer’s request for evidence of transferability, especially to new urban environments where some modalities may not be directly reusable.

---

> > > > ### Author Response · Authors · 2025-08-06
> > > >
> > > > Response to W6 and Q4: We thank the reviewer for highlighting fairness and efficiency concerns. Below we respond to each point:
> > > >
> > > > (1) Training Epochs
> > > >
> > > > We would like to clarify that this discrepancy was not intended to create an unfair comparison. In our experiments, each method was trained following the same hyperparameter settings provided in their papers or official implementations. Each method was trained until convergence based on validation performance, and different models exhibit different convergence behaviors.
> > > >
> > > > Our model operates under a multi-city, multi-view setting, which introduces greater data heterogeneity compared to single-city baselines. It requires more training steps to reach convergence. The decision to train SMARTraj^2 for 70 epochs is not arbitrary—it was guided by validation loss stabilization.
> > > >
> > > > To ensure fairness, we re-trained key baselines (e.g., MVTraj, JGRM) for up to 70 epochs:
> > > > (a) Their performance did not improve significantly beyond 30 epochs, confirming their early convergence.
> > > > (b) Training SMARTraj^2 for only 30 epochs led to notable performance drops, underscoring need for extended training under more complex settings.
> > > >
> > > > Since figures are not permitted in the rebuttal, we will provide convergence curves in the final version.
> > > >
> > > > (2) Efficiency Trade-off and Broader Improvements
> > > >
> > > > We recognize that SMARTraj^2 introduces higher computational cost due to its multi-view fusion and cross-city modeling. However, this design enables stronger generalization across diverse urban environments, which is the primary goal of our work.
> > > >
> > > > While SMARTraj^2 has a longer inference time (11.65ms vs. 4.22ms), it achieves consistently stronger performance across all tasks (Table 1 in Sec 4.2, and Table 4 in Appendix B.2.1).
> > > >
> > > > Moreover, this advantage becomes more pronounced in low-resource or cross-city scenarios. Unlike JGRM, which must be trained from scratch for each city, SMARTraj^2 can be pretrained on multiple source cities and then efficiently fine-tuned on a new city using limited data, thanks to its transferable representations. This not only improves generalization but also significantly reduces total cost of training and data annotation when scaling to multiple cities.
> > > >
> > > > We believe this is a reasonable and deliberate trade-off in scenarios where cross-city adaptability and robust generalization are more critical than marginal inference latency — such as nationwide logistics, urban planning tools, or multi-city ride-sharing platforms.
> > > >
> > > > Nonetheless, we acknowledge that inference efficiency is important in latency-sensitive applications. We will include this limitation, along with future optimization directions (e.g., model distillation or lightweight architectures), in a dedicated "Limitation and Future Work" section.
> > > >
> > > > (3) Performance Attribution: GPS vs. Architecture
> > > >
> > > > We agree that GPS view offers rich information. However, our ablation study demonstrates that the performance gain is not solely from GPS inputs:
> > > >
> > > > | Method                     | Road Label      |                         | Travel Time |                         | Destination Road |                         | Destination Grid |                         |
> > > > |-----------------------------|------------|-------------|----------|-----------|---------|----------|----------|----------|
> > > > |                             | Micro-F1 ↑ | Macro-F1 ↑ | MAE ↓    | RMSE ↓   | Acc@1 ↑ | Acc@5 ↑ | Acc@1 ↑ | Acc@5 ↑|
> > > > | SMARTraj^2                  | 0.8407     | 0.8298      | 35.0689  | 60.9156   | 0.7409  | 0.9069   | 0.6675   | 0.8392   |
> > > > | w/o grid                    | 0.8233     | 0.8186      | 72.6226  | 105.6123  | 0.6604  | 0.8402   | X        | X        |
> > > > | w/o GPS                     | 0.7987     | 0.7832      | 73.2965  | 106.1142  | 0.5446  | 0.7667   | 0.4110   | 0.6351   |
> > > > | w/o route                   | X          | X           | 74.5902  | 106.7897  | 0.5924  | 0.8049   | 0.4311   | 0.6636   |
> > > > | w/o gating                  | 0.8387     | 0.8279      | 40.3469  | 68.6216   | 0.6787  | 0.8478   | 0.5244   | 0.7200   |
> > > > | w/o invariant+specific      | 0.7415     | 0.7268      | 56.5380  | 71.5127   | 0.4770  | 0.7001   | 0.2556   | 0.5315   |
> > > > | w/o gating + specific       | 0.7637     | 0.7574      | 54.8681  | 84.8157   | 0.6300  | 0.8294   | 0.5466   | 0.7636   |
> > > >
> > > > This confirms that architectural components are critical to the improvements, and the performance cannot be attributed to GPS input alone.
> > > >
> > > > (4) Clarifying Our Model Design Philosophy
> > > >
> > > > SMARTraj^2 is not intended as a lightweight or efficiency-oriented model, but rather as a generalizable, scalable solution for multi-city trajectory modeling. Its modular architecture is specifically designed to disentangle and adapt to diverse urban structures. We believe this approach is valuable for real-world applications where transferability and scalability outweigh marginal efficiency costs. Our contribution lies in this flexible and stable design, beyond longer training or additional inputs.

---

> > > > ### Author Response · Authors · 2025-08-09
> > > >
> > > > We appreciate the reviewer’s insightful suggestion regarding the need to evaluate the generalization ability of our approach to cities with different spatial structures.
> > > >
> > > > To address this concern, we have initiated additional experiments on the Porto dataset, which features a decentralized European-style road network. These experiments follow exactly the same model configuration and evaluation protocols as in our main study, thereby enabling a direct comparison.
> > > >
> > > > Due to the time required for Porto data processing and GPU sharing constraints, our three-city training (Xi’an + Chengdu + Porto) has so far only completed 15 epochs and is therefore far from convergence, unlike the single-city and two-city settings where the model has fully converged. Nonetheless, the preliminary results already indicate that even at an early training stage, the three-city setting shows competitive performance compared with the converged two-city setting. We expect the performance to further improve once the training is complete.
> > > >
> > > > | Method                  | Road Label (Micro-F1 / Macro-F1) |
> > > > | ----------------------- | -------------------------------- |
> > > > | three-city (15th epoch) | 0.8373 / 0.8308                  |
> > > > | two-city (converged)    | 0.8407 / 0.8298                  |
> > > > | single-city (converged) | 0.8325 / 0.8144                  |
> > > >
> > > > These preliminary findings indicate that our approach can adapt to markedly different urban topologies and that incorporating a decentralized city such as Porto does not hinder — and likely enhances — the generalization capacity of the model.

---

### Official Review · Reviewer_fyeL · 2025-06-30

**Clarity:** 4
**Significance:** 4
**Originality:** 3
**Rating:** 5
**Confidence:** 5

**Summary:**

The paper introduces a novel method for multi-view spatio-temporal trajectory representation learning designed to address the challenges of generalizing across diverse urban scenes. The key contributions include: 1) Feature Disentanglement Module: Separates domain-invariant and domain-specific features using orthogonality constraints to handle multi-city structural heterogeneity. 2) Personalized Gating Mechanism: Dynamically adjusts contributions of different views and tasks to mitigate the amplified seesaw phenomenon in multi-city settings. 3) Experimental Validation: Demonstrates superior performance on benchmark datasets (Chengdu and Xi’an) across tasks like travel time estimation and destination prediction, outperforming state-of-the-art baselines.

**Questions:**

1) Does the computational cost of city-specific feature extractors become prohibitive as the number of cities increases?

2) The method assumes aligned multi-view data (GPS, route, grid). How does it perform if one or more views are missing or noisy (e.g., sparse GPS signals)?

3) Would the proposed method still work if only a single view (e.g., GPS-only) is available in some cities?

4) The method is evaluated on two cities (Chengdu and Xi’an). Could it scale to larger city networks (e.g., 10+ cities)?

**Ethical Concerns:**

["NO or VERY MINOR ethics concerns only"]

**Limitations:**

Suggestions for Improvement: Expand discussion on potential misuse (e.g., privacy risks from trajectory data) and safeguards (e.g., anonymization).

**Quality:**

3

**Strengths And Weaknesses:**

First to tackle multi-city, multi-view trajectory learning with explicit focus on generalization, addressing critical gaps in prior work (e.g., single-city limitations). And the feature disentanglement and gating mechanisms are well-designed, supported by ablation studies and theoretical grounding (e.g., orthogonality constraints). Additional discussion on the following aspects would enhance the paper:

1) Does the computational cost of city-specific feature extractors become prohibitive as the number of cities increases?

2) The method assumes aligned multi-view data (GPS, route, grid). How does it perform if one or more views are missing or noisy (e.g., sparse GPS signals)?

3) Would the proposed method still work if only a single view (e.g., GPS-only) is available in some cities?

4) The method is evaluated on two cities (Chengdu and Xi’an). Could it scale to larger city networks (e.g., 10+ cities)?

---

> ### Author Rebuttal · Authors · 2025-07-30
>
> We thank Reviewer fyeL for the insightful comments. We address each point below:
>
> **W1\&Q1**. We conducted additional experiments to evaluate computational cost and scalability. As shown in the table below, our method maintains a reasonable trade-off between model size and performance, even with the city-specific components. Details and further analysis will be included in the revised manuscript.
> | Method       | Model Size (MB) | Train Time (min/epoch) | Inference Time (ms) |
> |--------------|----------------:|-----------------------:|--------------------:|
> | Random       | -               | -                      | 0.374               |
> | Word2Vec     | 8.0             | 0.2                    | 0.404               |
> | Node2Vec     | 7.6             | 0.2                    | 0.328               |
> | Transformer  | 14              | 1.5                    | 0.940               |
> | BERT         | 433             | 3.5                    | 1.260               |
> | Toast        | 27              | 7.2                    | 2.152               |
> | JCLRNT       | 13              | 1.3                    | 0.792               |
> | START        | 94              | 17                     | 4.700               |
> | JGRM         | 33              | 15                     | 4.220               |
> | MVTraj       | 207             | 35                     | 10.560              |
> | Ours         | 477             | 54                     | 11.65               |
> ***
> **W2\&Q2**. We acknowledge that the route view is city-specific due to locally defined road segment IDs, which limits its transferability. However, other views (GPS, POI, grid) are transferable across cities when similar raw data is available. Our model is designed to function effectively using only the transferable views, and we will elaborate on this in the revised manuscript.
> ***
> **W3\&Q3**. We conducted an ablation study to evaluate the model under limited-view settings. The results below demonstrate that our method maintains reasonable performance even when only GPS data is available.
> | Method   | Road Label Classification |           | Travel Time Estimation |           | Destination Road Prediction |           | Destination Grid Prediction |           |
> |----------|---------------------------|-----------|------------------------|-----------|-----------------------------|-----------|-----------------------------|-----------|
> |          | Micro-F1 ↑                | Macro-F1 ↑| MAE ↓                  | RMSE ↓    | Acc@1 ↑                     | Acc@5 ↑   | Acc@1 ↑                     | Acc@5 ↑   |
> | Ours     | 0.8407                    | 0.8298    | 35.0689                | 60.9156   | 0.7409                      | 0.9069    | 0.6675                      | 0.8392    |
> | w/o grid | 0.8233                    | 0.8186    | 72.6226                | 105.6123  | 0.6604                      | 0.8402    | X                           | X         |
> | w/o GPS  | 0.7987                    | 0.7832    | 73.2965                | 106.1142  | 0.5446                      | 0.7667    | 0.4110                      | 0.6351    |
> | w/o route| X                         | X         | 74.5902                | 106.7897  | 0.5924                      | 0.8049    | 0.4311                      | 0.6636    |
> ***
> **W4\&Q4**. Although current experiments involve only two cities (Chengdu and Xi’an), our framework is designed to scale across multiple city domains. The dual-channel architecture (shared and city-specific) and personalized gating mechanism are inherently scalable without requiring architectural modifications. Future experiments on larger city networks will be explored and discussed in the revised manuscript.
> ***
> **L1**. We will expand the manuscript to explicitly discuss ethical and societal considerations. Specifically: (1) A dedicated 'Limitations' subsection will address multi-view data dependency, non-transferable components, and potential domain shift issues. (2) The 'Societal Impact' section will be extended to discuss potential risks (e.g., privacy concerns from trajectory data), as well as safeguards such as data anonymization and aggregation.

---

> > ### Comment · Reviewer_fyeL · 2025-08-05
> >
> > I appreciate the authors' responses. My concerns have been addressed, and I am satisfied with the clarifications provided. I maintain my positive score.

---

> > > ### Author Response · Authors · 2025-08-06
> > >
> > > Thank you very much for your positive assessment and for taking the time to read our rebuttal. We're glad to hear that our responses addressed your concerns. We truly appreciate your thoughtful review.

---

### Decision · Program_Chairs · 2025-09-17

**Decision:**

Accept (poster)

**Comment:**

The paper received 3 accept and 1 reject ratings. Reviewer praised for the approach of multiple-city modeling while concerning about the fairness of experiments, as well as the presentation. The authors provided a rebuttal. Most of those concerns were addressed while reviewers still have concerns, regarding the number of training epochs for baselines (fewer epochs). AC tends to trust the authors that those baselines, since they are single city based modeling, trains faster and converges within fewer epochs. AC recommends to accept this paper. Please incorporate the suggestions given by reviewers, especially  for the number of epochs in baseline experiments.  If it converges within fewer epochs, please still train them with more epochs so the audiences can see the performance is truly saturated.